# A First-Occupancy Representation for Reinforcement Learning

**Ted Moskovitz**[1]* **Spencer R. Wilson**[2] **Maneesh Sahani**[1]
[1]Gatsby Unit, UCL [2] Sainsbury Wellcome Centre, UCL

## Abstract

Both animals and artificial agents benefit from state representations that support rapid transfer of learning across tasks and which enable them to efficiently traverse their environments to reach rewarding states. The *successor representation* (SR), which measures the expected cumulative, discounted state occupancy under a fixed policy, enables efficient transfer to different reward structures in an otherwise constant Markovian environment and has been hypothesized to underlie aspects of biological behavior and neural activity. However, in the real world, rewards may only be available for consumption once, may shift location, or agents may simply aim to reach goal states as rapidly as possible without the constraint of artificially imposed task horizons. In such cases, the most behaviorally-relevant representation would carry information about when the agent was likely to *first* reach states of interest, rather than how often it should expect to visit them over a potentially infinite time span. To reflect such demands, we introduce the *first-occupancy representation* (FR), which measures the expected temporal discount to the first time a state is accessed. We demonstrate that the FR facilitates exploration, the selection of efficient paths to desired states, allows the agent, under certain conditions, to plan provably optimal trajectories defined by a sequence of subgoals, and induces similar behavior to animals avoiding threatening stimuli.

## 1 Introduction

In order to maximize reward, both animals and machines must quickly make decisions in uncertain environments with rapidly changing reward structure. Often, the strategies these agents employ are categorized as either model-free (MF) or model-based (MB) (Sutton & Barto, 2018). In the former, the optimal action in each state is identified through trial and error, with propagation of learnt value from state to state. By contrast, the latter depends on the acquisition of a map-like representation of the environment's transition structure, from which an optimal course of action may be derived.

This dichotomy has motivated a search for intermediate models which cache information about environmental structure, and so enable efficient but flexible planning. One such approach, based on the *successor representation* (SR) (Dayan, 1993), has been the subject of recent interest in the context of both biological (Stachenfeld et al., 2017; Gershman, 2018; Momennejad et al., 2017; Vértes & Sahani, 2019; Behrens et al., 2018) and machine (Kulkarni et al., 2016; Barreto et al., 2017b;a; 2018; Machado et al., 2020; Ma et al., 2020; Madarasz & Behrens, 2019) learning. The SR associates with each state and policy of action a measure of the expected rate of future occupancy of all states if that policy were to be followed indefinitely. This cached representation can be acquired through experience in much the same way as MF methods and provides some of the flexibility of MB behaviour at reduced computational cost. Importantly, the SR makes it possible to rapidly evaluate the expected return of each available policy in an otherwise unchanging environment, provided that the transition distribution remains consistent.

However, these requirements limit the applicability of the SR. In the real world, rewards are frequently non-Markovian. They may be depleted by consumption, frequently only being available on the first entry to a state. Internal goals for control—say, to pick up a particular object—need to be achieved as rapidly as possible, but only once at a time.

---

*Correspondence: `ted@gatsby.ucl.ac.uk`

Furthermore, while a collection of SRs for different policies makes it possible to select the best amongst them, or to improve upon them all by considering the best immediate policy-dependent state values (Barreto et al., 2018), this capacity still falls far short of the power of planning within complete models of the environment.

Here, we propose a different form of representation in which the information cached is appropriate for achieving ephemeral rewards and for planning complex combinations of policies. Both features arise from considering the expected time at which other states will be *first* accessed by following the available policies. We refer to this as a *first-occupancy representation* (FR). The shift from expected rate of future occupancy (SR) to delay to first occupancy makes it possible to handle settings where the underlying environment remains stationary, but reward availability is not Markovian. *Our primary goal in this paper is to formally introduce the FR and to highlight the breadth of settings in which it offers a compelling alternative to the SR, including, but not limited to: exploration, unsupervised RL, planning, and modeling animal behavior.*

## 2 REINFORCEMENT LEARNING PRELIMINARIES

**Policy evaluation and improvement**    In reinforcement learning (RL), the goal of the agent is to act so as to maximize the discounted cumulative reward received within a task-defined environment. We model a task $T$ as a finite *Markov decision process* (MDP; (Puterman, 2010)), $T = (\mathcal{S}, \mathcal{A}, p, r, \gamma, \mu)$, where $\mathcal{S}$ is a finite state space, $\mathcal{A}$ is a finite action space, $p : \mathcal{S} \times \mathcal{A} \to \Delta(\mathcal{S})$ is the transition distribution (where $\Delta(\mathcal{S})$ is the probability simplex over $\mathcal{S}$), $r : \mathcal{S} \to \mathbb{R}$ is the reward function, $\gamma \in [0, 1)$ is a discount factor, and $\mu \in \Delta(\mathcal{S})$ is the distribution over initial states.

Note that the reward function is also frequently defined over state-action pairs $(s, a)$ or triples $(s, a, s')$, but we restrict our analysis to state-based rewards for now. The goal of the agent is to maximize its expected *return*, or discounted cumulative reward $\sum_t \gamma^t r(s_t)$. To simplify notation, we will frequently write $r(s_t) := r_t$ and $\mathbf{r} \in \mathbb{R}^{|\mathcal{S}|}$ as the vector of rewards for each state. The agent acts according to a stationary policy $\pi : \mathcal{S} \to \Delta(\mathcal{A})$. For finite MDPs, we can describe the expected transition probabilities under $\pi$ using a $|\mathcal{S}| \times |\mathcal{S}|$ matrix $P^\pi$ such that $P^\pi_{s,s'} = p^\pi(s'|s) := \sum_a p(s'|s, a)\pi(a|s)$. Given $\pi$ and a reward function $r$, the expected return is

$$Q_r^\pi(s, a) = \mathbb{E}_\pi \left[ \sum_{k=0}^\infty \gamma^k r_{t+k} \Big| s_t = s, a_t = a \right] = \mathbb{E}_{\substack{s' \sim p^\pi(\cdot|s) \\ a' \sim \pi(\cdot|s')}} \left[ r_t + \gamma Q_r^\pi(s', a') \right]. \qquad (1)$$

$Q_r^\pi$ are called the state-action values or simply the $Q$-values of $\pi$. The expectation $\mathbb{E}_\pi [\cdot]$ is taken with respect to the randomness of both the policy and the transition dynamics. For simplicity of notation, from here onwards we will write expectations of the form $\mathbb{E}_\pi [\cdot|s_t = s, a_t = a]$ as $\mathbb{E}_\pi [\cdot|s_t, a_t]$. This recursive form is called the *Bellman equation*, and it makes the process of estimating $Q_r^\pi$—termed *policy evaluation*—tractable via dynamic programming (DP; Bellman, 1957). In particular, successive applications of the *Bellman operator* $\mathcal{T}^\pi Q := r + \gamma P^\pi Q$ are guaranteed to converge to the true value function $Q^\pi$ for any initial real-valued $|\mathcal{S}| \times |\mathcal{A}|$ matrix $Q$.

When the transition dynamics and reward function are unknown, *temporal difference* (TD) learning updates value estimates using a bootstrapped estimate of the Bellman target (Sutton & Barto, 2018). Given a transition sequence $(s_t, a_t, r_t, s_{t+1})$ and $a_{t+1} \sim \pi(\cdot|s_{t+1})$,

$$Q_r^\pi(s_t, a_t) \leftarrow Q_r^\pi(s_t, a_t) + \alpha\delta_t, \qquad \delta_t := r_t + \gamma Q_r^\pi(s_{t+1}, a_{t+1}) - Q_r^\pi(s_t, a_t). \qquad (2)$$

Once a policy has been evaluated, *policy improvement* identifies a new policy $\pi'$ such that $Q_r^\pi(s, a) \geq Q_r^{\pi'}(s, a)$, $\forall (s, a) \in Q_r^\pi(s, a)$. Helpfully, such a policy can be defined as $\pi'(s) \in \operatorname{argmax}_a Q_r^\pi(s, a)$.

**The successor representation**    The *successor representation* (SR; (Dayan, 1993)) is motivated by the idea that a state representation for policy evaluation should be dependent on the similarity of different paths under the current policy. The SR is a policy's expected cumulative discounted state occupancy, and for discrete state spaces can be stored in an $|\mathcal{S}| \times |\mathcal{S}|$ matrix $M^\pi$, where

$$M^\pi(s, s') := \mathbb{E}_\pi \left[ \sum_{k=0}^\infty \gamma^k \mathbb{1}(s_{t+k} = s') \Big| s_t \right] = \mathbb{E}_\pi \left[ \mathbb{1}(s_t = s') + \gamma M^\pi(s_{t+1}, s') \Big| s_t \right], \qquad (3)$$

where $\mathbb{1}(\cdot)$ is the indicator function. The SR can also be conditioned on actions, i.e., $M^\pi(s,a,s') :=$ $\mathbb{E}_\pi \left[ \sum_k \gamma^k \mathbb{1}(s_{t+k} = s') \middle| s_t, a_t \right]$, and expressed in a vectorized format, we can write $M^\pi(s) :=$ $M^\pi(s, \cdot)$ or $M^\pi(s,a) := M^\pi(s,a,\cdot)$. The recursion in Eq. (3) admits a TD error:

$$\delta_t^M := \mathbf{1}(s_t) + \gamma M^\pi(s_{t+1}, \pi(s_{t+1})) - M^\pi(s_t, a_t), \tag{4}$$

where $\mathbf{1}(s_t)$ is a one-hot state representation of length $|\mathcal{S}|$. One useful property of the SR is that, once converged, it facilitates rapid policy evaluation for any reward function in a given environment:

$$\mathbf{r}^\mathsf{T} M^\pi(s,a) = \mathbf{r}^\mathsf{T} \mathbb{E}_\pi \left[ \sum_k \gamma^k \mathbb{1}(s_{t+k}) \middle| s_t, a_t \right] = \mathbb{E}_\pi \left[ \sum_k \gamma^k r_{t+k} \middle| s_t, a_t \right] = Q_r^\pi(s,a). \tag{5}$$

**Fast transfer for multiple tasks** In the real world, we often have to perform multiple tasks within a single environment. A simplified framework for this scenario is to consider a set of MDPs $\mathcal{M}$ that share every property (i.e., $\mathcal{S}, \mathcal{A}, p, \gamma, \mu$) except reward functions, where each task within this family is determined by a reward function $r$ belonging to a set $\mathcal{R}$. Extending the notions of policy evaluation and improvement to this multitask setting, we can define *generalized policy evaluation* (GPE) as the computation of the value function of a policy $\pi$ on a set of tasks $\mathcal{R}$. Similarly, *generalized policy improvement* (GPI) for a set of "base" policies $\Pi$ is the definition of a policy $\pi'$ such that

$$Q_r^{\pi'}(s,a) \geq \sup_{\pi \in \Pi} Q_r^\pi(s,a) \; \forall (s,a) \in \mathcal{S} \times \mathcal{A} \tag{6}$$

for some $r \in \mathcal{R}$. As hinted above, the SR offers a way to take advantage of this shared structure by decoupling the agent's evaluation of its expected transition dynamics under a given policy from a single reward function. Rather than needing to directly estimate $Q^\pi \; \forall \pi \in \Pi$, $M^\pi$ only needs to be computed once, and given a new reward vector $\mathbf{r}$, the agent can quickly peform GPE via Eq. (5). As shown by Barreto et al. (2017a), GPE and GPI can be combined to define a new policy $\pi'$ via

$$\pi'(s) \in \operatorname*{argmax}_{a \in \mathcal{A}} \max_{\pi \in \Pi} \mathbf{r}^\mathsf{T} M^\pi(s,a). \tag{7}$$

For brevity, we will refer to this combined procedure of GPE and GPI simply as "GPI", unless otherwise noted. The resulting policy $\pi'$ is guaranteed to perform at least as well as any individual $\pi \in \Pi$ (Barreto et al., 2020) and is part of a larger class of policies termed *set-improving policies* which perform at least as well as any single policy in a given set (Zahavy et al., 2021).

## 3 THE FIRST-OCCUPANCY REPRESENTATION

While the SR encodes states via total occupancy, this may not always be ideal. If a task lacks a time limit but terminates once the agent reaches a pre-defined goal, or if reward in a given state is consumed or made otherwise unavailable once encountered, a more useful representation would instead measure the duration until a policy is expected to reach states the *first* time. Such natural problems emphasize the importance of the first occupancy and motivate the *first-occupancy representation* (FR).

**Definition 3.1.** *For an MDP with finite $\mathcal{S}$, the first-occupancy representation (FR) for a policy $\pi$ $F^\pi \in [0,1]^{|\mathcal{S}| \times |\mathcal{S}|}$ is given by*

$$F^\pi(s,s') := \mathbb{E}_\pi \left[ \sum_{k=0}^\infty \gamma^k \mathbb{1}(s_{t+k} = s', s' \notin \{s_{t:t+k}\}) \middle| s_t \right], \tag{8}$$

*where $\{s_{t:t+k}\} = \{s_t, s_{t+1}, \ldots, s_{t+k-1}\}$, with the convention that $\{s_{t:t+0}\} = \varnothing$.*

That is, as the indicator equals 1 iff $s_{t+k} = s'$ *and* time $t+k$ is the first occasion on which the agent has occupied $s'$ since time $t$, $F^\pi(s,s')$ gives the expected discount at the time the policy first reaches $s'$ starting from $s$. The idea of learning policy-dependent distances to target states has a long history in RL (Kaelbling, 1993; Pong et al., 2018; Hartikainen et al., 2020). However, previous methods don't learn these distances as state *representations* and measure distance in the space of time steps, rather than discount factors. A more thorough discussion can be found in Appendix A.8. We can write a recursive relationship for the FR (derivation in Appendix A.1):

$$F^\pi(s,s') = \mathbb{E}_{s_{t+1} \sim p^\pi(\cdot|s)} \left[ \mathbb{1}(s_t = s') + \gamma(1 - \mathbb{1}(s_t = s')) F^\pi(s_{t+1}, s') \middle| s_t \right] \tag{9}$$

This recursion implies the following Bellman operator, analogous to the one used for policy evaluation:

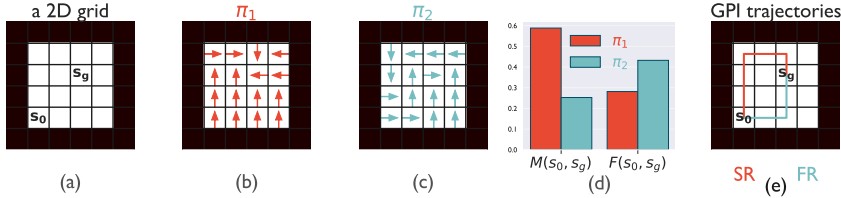

Figure 1: **The FR is higher for shorter paths.** (a-c) A 2D gridworld and fixed policies. (d) The FR from $s_0$ to $s_g$ is higher for $\pi_2$, but the SR is lower. (e) SR-GPI with the SR picks $\pi_1$, while FR-GPI selects $\pi_2$.

**Definition 3.2** (FR Operator). *Let $F \in \mathbb{R}^{|\mathcal{S}| \times |\mathcal{S}|}$ be an arbitrary real-valued matrix. Then let $\mathcal{G}^\pi$ denote the Bellman operator for the FR, such that*

$$\mathcal{G}^\pi F = I_{|\mathcal{S}|} + \gamma(\mathbf{1}\mathbf{1}^\mathsf{T} - I_{|\mathcal{S}|})P^\pi F, \tag{10}$$

*where $\mathbf{1}$ is the length-$|\mathcal{S}|$ vector of all ones. In particular, for a stationary policy $\pi$, $\mathcal{G}^\pi F^\pi = F^\pi$.*

The following result establishes $\mathcal{G}^\pi$ as a contraction, with the proof provided in Appendix A.4.

**Proposition 3.1** (Contraction). *Let $\mathcal{G}^\pi$ be the operator as defined in Definition 3.2 for some stationary policy $\pi$. Then for any two matrices $F, F' \in \mathbb{R}^{|\mathcal{S}| \times |\mathcal{S}|}$,*

$$|\mathcal{G}^\pi F(s, s') - \mathcal{G}^\pi F'(s, s')| \leq \gamma|F(s, s') - F'(s, s')|, \tag{11}$$

*with the difference equal to zero for $s = s'$.*

This implies the following convergence property of $\mathcal{G}^\pi$.

**Proposition 3.2** (Convergence). *Under the conditions assumed above, set $F^{(0)} = I_{|\mathcal{S}|}$. For $k = 0, 1, \ldots$, suppose $F^{(k+1)} = \mathcal{G}^\pi F^{(k)}$. Then*

$$|F^{(k)}(s, s') - F^\pi(s, s')| < \gamma^k \tag{12}$$

*for $s \neq s'$ with the difference for $s = s'$ equal to zero $\forall k$.*

Therefore, repeated applications of the FR Bellman operator $\mathcal{G}^k F \to F^\pi$ as $k \to \infty$. When the transition matrix $P^\pi$ is unknown, the FR can instead be updated through the following TD error:

$$\delta_t^F = \mathbb{1}(s_t = s') + \gamma(1 - \mathbb{1}(s_t = s'))F^\pi(s_{t+1}, s') - F^\pi(s_t, s'). \tag{13}$$

In all following experiments, the FR is learned via TD updates, rather than via dynamic programming. To gain intuition for the FR, we can imagine a 2D environment with start state $s_0$, a rewarded state $s_g$, and deterministic transitions (Fig. 1a). One policy, $\pi_1$, reaches $s_g$ slowly, but after first encountering it, re-enters $s_g$ infinitely often (Fig. 1b). A second policy, $\pi_2$, reaches $s_g$ quickly but never occupies it again (Fig. 1c). In this setting, because $\pi_1$ re-enters $s_g$ multiple times, despite arriving there more slowly than $\pi_2$, $M^{\pi_1}(s_0, s_g) > M^{\pi_2}(s_0, s_g)$, but because the FR only counts the first occupancy of a given state, $F^{\pi_1}(s_0, s_g) < F^{\pi_2}(s_0, s_g)$. The FR thus reflects a policy's path length between states.

**Policy evaluation and improvement with the FR** As with the SR, we can quickly perform policy evaluation with the FR. Crucially, however, the FR induces the following value function:

$$\mathbf{r}^\mathsf{T} F^\pi(s, a) = \mathbb{E}_\pi\left[\sum_k \gamma^k r_{t+k}^F \Big| s_t, a_t\right] := Q_{r^F}^\pi(s, a), \tag{14}$$

where $r^F : \mathcal{S} \to \mathbb{R}$ is a reward function such that $r^F(s_{t+k}) = r(s_{t+k})$ if $s_{t+k} \notin \{s_{t:t+k}\}$ and 0 otherwise. In other words, multiplying any reward vector by the FR results in the value function for a corresponding task with *non-Markovian* reward structure in which the agent obtains rewards from states only once. Policy improvement can then be performed with respect to $Q_{r^F}^\pi$ as normal. This structure is a very common feature of real-world tasks, such as foraging for food or reaching to a target. Accordingly, there is a rich history of studying tasks with this kind of structure, termed *non-Markovian reward decision processes* (NMRDPs; (Peshkin et al., 2001; Littman et al., 2017;

Gaon & Brafman, 2020; Ringstrom & Schrater, 2019)). Helpfully, all NMRDPs can be converted into an equivalent MDP with an appropriate transformation of the state space (Bacchus et al., 1996).

Most approaches in this family attempt to be generalists, *learning* an appropriate state transformation and encoding it using some form of logical calculus or finite state automaton (Bacchus et al., 1996; Littman et al., 2017; Gaon & Brafman, 2020). While it would technically be possible to learn or construct the transformation required to account for the non-Markovian nature of $r^F$, it would be exponentially expensive in $|\mathcal{S}|$, as every path would need to account for the first occupancy of each state along the path. That is, $|\mathcal{S}|$ bits would need to be added to each successive state in the trajectory. Crucially, the FR has the added advantage of being task-agnostic, in that for *any* reward function $r$ in a given environment, the FR can immediately perform policy evaluation for the corresponding $r^F$.

**Infinite state spaces** A natural question when extending the FR to real-world scenarios is how it can be generalized to settings where $|\mathcal{S}|$ is either impractically large or infinite[1]. In these cases, the SR is reframed as *successor features* $\psi^\pi$ (SFs; Kulkarni et al., 2016; Barreto et al., 2017b), where the $d$th SF is defined as $\psi_d^\pi(s) := \mathbb{E}_\pi\left[\sum_{k=0}^\infty \gamma^k \phi_d(s_{t+k}) \big| s_t = s\right]$, where $d = 1, \ldots, D$ and $\phi : \mathcal{S} \to \mathbb{R}^D$ is a *base feature* function. The base features $\phi(\cdot)$ are typically defined so that a linear combination predicts immediate reward (i.e., $\mathbf{w}^\top \phi(s) = r(s)$ for some $\mathbf{w} \in \mathbb{R}^D$), and there are a number of approaches to learning them (Kulkarni et al., 2016; Barreto et al., 2018; Ma et al., 2020). A natural extension to continuous $\mathcal{S}$ for the FR would be to define a *first-occupancy feature* (FF) representation $\varphi^\pi$, where the $d$th FF is given by

$$\varphi_d^\pi(s) := \mathbb{E}_\pi\left[\sum_{k=0}^\infty \gamma^k \mathbb{1}(\phi_d(s_{t+k}) \geq \theta_d, \{\phi_d(s_{t'})\}_{t'=t:t+k} < \theta_d) \Big| s_t = s\right] \tag{15}$$
$$= \mathbb{1}(\phi_d(s_t) \geq \theta_d) + \gamma(1 - \mathbb{1}(\phi_d(s_t) \geq \theta_d))\mathbb{E}_{s_{t+1} \sim p^\pi}[\varphi_d^\pi(s_{t+1})]$$

where $\theta_d$ is a threshold value for the $d$th feature. The indicator equals 1 only if $s_{t+k}$ is the first state whose feature embedding exceeds the threshold. Note that this representation recovers the FR when the feature function is a one-hot state encoding and the thresholds $\{\theta_d\}_{d=1}^D$ are all 1.

## 4 EXPERIMENTS

We now demonstrate the broad applicability of the FR, and highlight ways its properties differ from those of the SR. We focus on 4 areas: exploration, unsupervised RL, planning, and animal behavior.

### 4.1 THE FR AS AN EXPLORATION BONUS

Intuitively, representations which encode state visitation should be useful measures of exploration. Machado et al. (2020) proposed the SR as a way to encourage exploration in tasks with sparse or deceptive rewards. Specifically, the SR is used as a bonus in on-policy learning with Sarsa (Rummery & Niranjan, 1994):

Table 1: Exploration results. $\pm$ values denote 1 SE across 100 trials.

| method | RIVERSWIM | SIXARMS |
|---|---|---|
| SARSA + FR | $1,547,243 \pm 34,050$ | $11,9149 \pm 42,942$ |
| SARSA + SR | $1,197,075 \pm 36,999$ | $1,025,750 \pm 49,095$ |
| SARSA | $25,075 \pm 1,224$ | $376,655 \pm 8,449$ |

$$\delta_t = r_t + \frac{\beta}{\|M^\pi(s_t)\|_1} + \gamma Q^\pi(s_{t+1}, \pi(s_{t+1})) - Q^\pi(s_t, a_t), \tag{16}$$

where $\beta \in \mathbb{R}$ controls the size of the exploration bonus. Machado et al. (2020) show that during learning, $\|M^\pi(s)\|_1$ can act as a proxy for the state-visit count $n(s)$, with $\|M^\pi(s)\|_1^{-1}$ awarding a progressively lower bonus for every consecutive visit of $s$. In the limit as $t \to \infty$, however, $\|M^\pi(s)\|_1^{-1} \to 1 - \gamma \ \forall s$ as $\pi$ stabilizes, regardless of whether $\pi$ has effectively explored. To encourage exploration, we'd like for a bonus to maintain its effectiveness over time. In contrast to $\|M^\pi(s)\|_1$, $1 \leq \|F^\pi(s)\|_1 \leq \kappa_{|\mathcal{S}|} := \frac{1-\gamma^{|\mathcal{S}|+1}}{1-\gamma}$, where $\kappa_{|\mathcal{S}|} > 1$ for $|\mathcal{S}| \geq 1$. Note that $\|F^\pi(s)\|_1 = \kappa_{|\mathcal{S}|}$ only if $\pi$ reaches all states in as many steps. Because $\|F^\pi\|_1$ only grows when new states or shorter paths are discovered, we can instead augment the reward as follows:

$$r_t \leftarrow r_t + \beta\|F^\pi(s_t)\|_1. \tag{17}$$

---

[1]Recent work (Blier et al., 2021; Touati & Ollivier, 2021) has shown ways of extending the SR to continuous $\mathcal{S}$ with the need for base features. We leave this as an interesting avenue for future work.

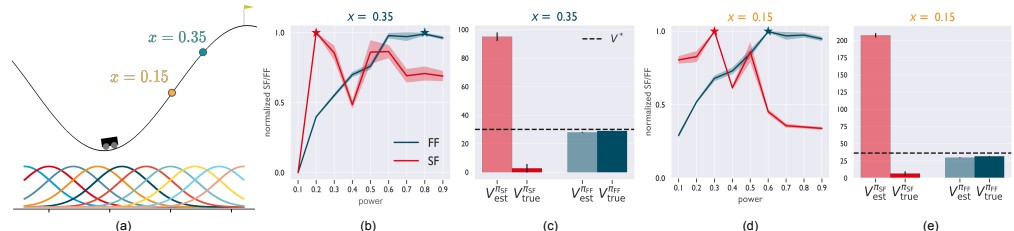

Figure 2: **The FF facilitates accurate policy evaluation and selection.** Shading denotes 1 SE over 20 seeds.

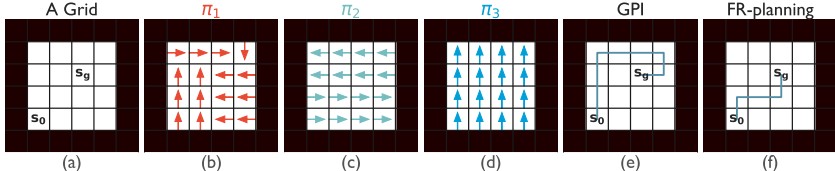

Figure 3: **The FR enables efficient planning.** (a-d) A 2D gridworld with start ($s_0$) and goal($s_g$) states, along with three fixed policies. (e) GPI follows $\pi_1$. (f) Planning with the FR enables the construction of a shorter path.

We tested our approach on the RIVERSWIM and SIXARMS problems (Strehl & Littman, 2008), two hard-exploration tasks from the PAC-MDP literature. In both tasks, visualized in Appendix Fig. 8, the transition dynamics push the agent towards small rewards in easy to reach states, with greater reward available in harder to reach states. In both cases, we ran Sarsa, Sarsa with an SR bonus (SARSA + SR) and Sarsa with an FR bonus (SARSA + FR) for 5,000 time steps with an $\epsilon$-greedy policy. The results are listed in Table 1, where we can see that the FR results in an added benefit over the SR. It's also important to note that the maximum bonus $\kappa_{|\mathcal{S}|}$ has another useful property, in that it scales exponentially in $|\mathcal{S}|$. This is desirable, because as the number of states grows, exploration frequently becomes more difficult. To measure this factor empirically, we tested the same approaches with the same settings on a modified RIVERSWIM, RIVERSWIM-N, with $N = \{6, 12, 24\}$ states, finding that SARSA + FR was more robust to the increased exploration difficulty (see Appendix Table 2 and Appendix A.3 for results and more details). We also tested whether these results extend to the function approximation setting, comparing a DQN-style model (Mnih et al., 2015) using the SF and FF as exploration bonuses on the challenging DEEPSEA task (Osband et al., 2020), finding that the advantage of the FF is conserved. See Appendix A.3 for details. Developing further understanding of the relationship between the FR and exploration represents an interesting topic for future work.

## 4.2 UNSUPERVISED RL WITH THE FF

We demonstrate the usefulness of the FF in the *unsupervised pre-training RL* (URL) setting, a paradigm which has gained popularity recently as a possible solution to the high sample complexity of deep RL algorithms (Liu & Abbeel, 2021; Gregor et al., 2016; Eysenbach et al., 2018; Sharma et al., 2020). In URL, the agent first explores an environment without extrinsic reward with the objective of learning a useful representation which then enables rapid fine-tuning to a test task.

**Continuous MountainCar** We first demonstrate that if the test task is non-Markovian, the SR can produce misleading value estimates. To do so, we use a modified version of the continuous MountainCar task (Brockman et al., 2016) (Fig. 2(a)). The agent pre-trains for 20,000 time steps in a rewardless environment, during which it learns FFs or SFs for a set of policies $\Pi$ which swing back and forth with a fixed acceleration or "power." (details in Appendix A.3). During fine-tuning, the agent must identify the policy $\pi \in \Pi$ which reaches a randomly sampled goal location the fastest. We use radial basis functions as the base features $\phi_d(\cdot)$ with fixed FF thresholds $\theta_d = 0.7$.

Fig. 2(b,d) plots the FF and SF values versus policy power from the start state for two different goal locations. The low-power policies require more time to gain momentum up the hill, but the policies which maximize the SF values slow down around the goal locations, dramatically increasing their total "occupancy" of that area. In contrast, high-powered policies reach the goal locations for the first time much sooner, and so the policies with the highest FF values have higher powers. In

the test phase, the agent fits the reward vector $\mathbf{w} \in \mathbb{R}^D$ by minimizing $\sum_t ||r_t - \mathbf{w}^\mathsf{T}\phi(s_t)||^2$, as is standard in the SF literature (Barreto et al., 2017b;a; 2018; 2020; Zahavy et al., 2021). The agent then follows the set-max policy (SMP; (Zahavy et al., 2021)), which selects the policy in $\Pi$ which has the highest expected value across starting states: $\pi^{\mathrm{SMP}} \in \operatorname{argmax}_{\pi \in \Pi} \mathbb{E}_{s_0 \sim \mu} [V^\pi(s_0)]$, where $V^\pi(s_0) = \mathbf{w}^\mathsf{T}\varphi^\pi(s_0)$ (with $\varphi^\pi$ replaced by $\psi^\pi$ for SF-based value estimates). Fig. 2(c,e) shows both the estimated ($V_{\mathrm{est}}$) and true ($V_{\mathrm{true}}$) values of the SMPs selected using both the SF and FF, along with the value of the optimal policy $V^*$. We can see that the accumulation of the SFs results in a significant overestimation in value, as well as a suboptimal policy. The FF estimates are nearly matched to the true values of the selected policies for each goal location and achieve nearly optimal performance.

**Robotic reaching**    To test whether these results translate to high-dimensional problems, we applied the FF to the 6-DoF JACO robotic arm environment from Laskin et al. (2021). In this domain, the arm must quickly reach to different locations and perform simple object manipulations (Fig. 4(a)). We modify the *Active Pre-training with Successor features* (APS; Liu & Abbeel, 2021) URL algorithm, which leverages a nonparametric entropy maximization objective in conjunction with SFs during pre-training

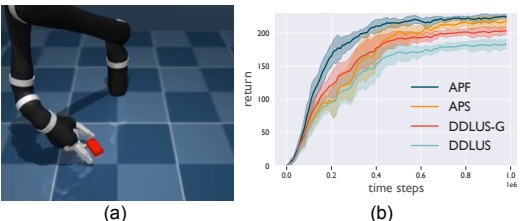

(a)                                              (b)

Figure 4: **APF accelerates convergence in robotic reaching.** Shading denotes 1 SE over 10 seeds.

(Hansen et al., 2020) in order to learn useful and adaptable behaviors. Our modification is to replace the SFs with FFs, resulting in *Active Pre-training with First-occupancy features* (APF), using the same intuition motivating the MountainCar experiments: cumulative features are misleading when downstream tasks benefit from quickly reaching a desired goal, in this case, the object. Here, the agent is first trained for $10^6$ time steps using the aforementioned intrinsic reward objective before being applied to a specified reaching task. As additional baselines, we compare against two variants of *dynamic distance learning - unsupervised* (DDLUS and DDLUS-G; (Hartikainen et al., 2020)). Details can be found in Appendix A.3. We found that the FF accelerated convergence (Fig. 4(b)).

### 4.3    PLANNING WITH THE FR

While SRs effectively encode task-agnostic, pre-compiled environment models, they cannot be directly used for multi-task model-based planning. GPI is only able to select actions based on a one-step lookahead, which may result in suboptimal behavior. One simple situation that highlights such a scenario is depicted in Fig. 3. As before, there are start and goal states in a simple room (Fig. 3(a)), but here there are three policies comprising $\Pi = \{\pi_1, \pi_2, \pi_3\}$ (Fig. 3(b-d)). GPI selects $\pi_1$ because it is the only policy that reaches the goal $s_g$ within one step of the start $s_0$: $\pi_1 = \max_{\pi \in \Pi} \mathbf{r}^\mathsf{T} M^\pi(s_0, s_g)$ (Fig. 3(e)). (Note that using GPI with the FR instead would also lead to this choice.) To gain further intuition for the FR versus the SR, we plot the representations for the policies in Appendix Fig. 17. However, the optimal strategy using the policies in $\Pi$ is instead to move right using $\pi_2$ and up using $\pi_3$. How can the FR be used to find such a sequence?

Intuitively, starting in a state $s$, this strategy is grounded in following one policy until a certain state $s'$, which we refer to as a *subgoal*, and then *switching* to a different policy in the base set. Because the FR effectively encodes the shortest path between each pair of states $(s, s')$ for a given policy, the agent can elect to follow the policy $\pi \in \Pi$ with the greatest value of $F^\pi(s, s')$, then switch to another policy and repeat the process until reaching a desired state. The resulting approach is related to the hierarchical *options* framework (Sutton et al., 1999; Sutton & Barto, 2018), where the planning process effectively converts the base policies into options with—as we show—optimal termination conditions. For a more detailed discussion, see Appendix A.7.

More formally, we can construct a DP algorithm to solve for the optimal sequence of planning policies $\pi^F$ and subgoals $s^F$. Denoting by $\Gamma_k(s)$ the total discount of the full trajectory from $s$ to $s_g$ at step $k$ of the procedure, we jointly optimize over policies $\pi$ and subgoals $s'$ for each state $s$:

$$\Gamma_{k+1}(s) = \max_{\pi \in \Pi, s' \in \mathcal{S}} F^\pi(s, s')\Gamma_k(s'), \quad \text{with} \quad \pi^F_{k+1}(s), s^F_{k+1}(s) = \operatorname*{argmax}_{\pi \in \Pi, s' \in \mathcal{S}} F^\pi(s, s')\Gamma_k(s').$$

Intuitively, the product $F^\pi(s, s')\Gamma_k(s')$ can be interpreted as the expected discount of the plan consisting of first following $\pi$ from $s$ to $s'$, then the current best (shortest-path) plan from $s'$ to $s_g$.

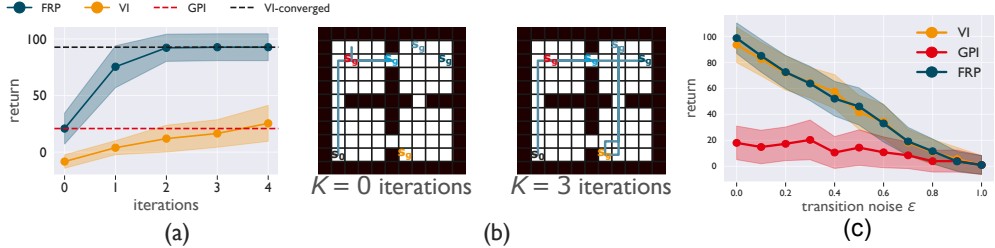

Figure 5: **FRP interpolates between GPI and model-based DP.** Shading represents 1 SE.

Note that it is this property of the FR which allows such planning: multiplying total occupancies, as would be done with the SR, is not well-defined. The full procedure, which we refer to as FR-planning (FRP), is given by Alg. 1 in Appendix A.2. Appendix Fig. 13 depicts the resulting policies $\pi^F$ and subgoals $s^F$ obtained from running FRP on the example in Fig. 3. The following result shows that under certain assumptions, FRP finds the shortest path to a given goal (proof in Appendix A.4).

**Proposition 4.1** (Planning optimality). *Consider a deterministic, finite MDP with a single goal state $s_g$, and a base policy set $\Pi$ composed of policies $\pi : \mathcal{S} \to \mathcal{A}$. We make the following coverage assumption: there exists a sequence of policies that reaches $s_g$ from a given start state $s_0$. Then Alg. 1 converges so that $\Gamma(s_0) = \gamma^{L_\pi^\star}$, where $L_\pi^\star$ is the shortest path length from $s_0$ to $s_g$ using $\pi \in \Pi$.*

**Performance and computational cost** We can see that each iteration of the planning algorithm adds at most one new subgoal to the planned trajectory from each state to the goal, with convergence when no policy switches can be made that reduce the number of steps required. If there are $K$ iterations, the overall computational complexity of FRP is $\mathcal{O}(K|\Pi||\mathcal{S}|^2)$. The worst-case complexity occurs when the policy must switch at every state en route to the target—$K$ is upper-bounded by the the number of states along the shortest path to the goal. In contrast, running value iteration (VI; (Sutton & Barto, 2018)) for $N$ iterations is $\mathcal{O}(N|\mathcal{A}||\mathcal{S}|^2)$. Given the true transition matrix $P$ and reward vector $\mathbf{r}$, VI will also converge to the shortest path to a specified goal state, but FRP converges more quickly than VI whenever $K|\Pi| < N|\mathcal{A}|$. To achieve an error $\epsilon$ between the estimated value function and the value function of the optimal policy, VI requires $N \geq \frac{1}{(1-\gamma)} \log \frac{2}{(1-\gamma)^2\epsilon}$ (Puterman, 1994), which for $\gamma = 0.95$, $\epsilon = 0.1$, e.g., gives $N \geq 180$ iterations[2]. To test convergence rates in practice, we applied FRP, VI, and GPI using the FR to the classic FOURROOMS environment (Sutton et al., 1999) on a modified task in which agents start in the bottom left corner and move to a randomly located goal state. Once the goal is reached, a new goal is randomly sampled in a different location until the episode is terminated after 75 time steps. For GPI and FRP, we use four base policies which each only take one action: {up, down, left, right}, with their FRs learned by TD learning. We ran each algorithm for 100 episodes, with the results plotted in Fig. 5(a). Note that here GPI is equivalent to FRP with $K = 0$ iterations. To see this, observe that when there is a single goal $s_g$ such that only $r(s_g) > 0$, the policy selected by GPI is

$$\pi^{\text{GPI}}(s) \in \operatorname*{argmax}_{\pi \in \Pi} \mathbf{r}^\mathsf{T} F^\pi(s) = \operatorname*{argmax}_{\pi \in \Pi} r(s_g) F^\pi(s, s_g) = \operatorname*{argmax}_{\pi \in \Pi} F^\pi(s, s_g). \tag{18}$$

When there are $n$ goal states with equal reward, finding the ordering of the goals that results in the shortest expected path is in general $\mathcal{O}(n!)$ (see Appendix A.6). Due to the nature of the base policies used above, the number of subgoals on any path is equal to the number of turns the agent must take from its curent state to the goal, which for this environment is three. We can then see that FRP reaches the optimal performance obtained by the converged VI after $K = 3$ iterations (Fig. 5(a)). In contrast, for the same number of iterations, VI performs far worse. This planning process must be repeated each time a new goal is sampled, so that the computational benefits of FRP versus traditional DP methods compound for each new reward vector. Example FRP trajectories between goals for $K = 0$ and $K = 3$ iterations are plotted in Fig. 5(b). Finally, to test FRP's robustness to stochasticity, we added transition noise $\epsilon$ to the FOURROOMS task. That is, the agent moves to a random adjacent state with probability $\epsilon$ regardless of action. We compared FRP to converged VI for increasing $\epsilon$, with the

---

[2]Other DP methods like policy iteration (PI), which is strongly polynomial, converge more quickly than VI, but in the example above, PI still needs $N \geq \log(1/((1 - \gamma)\epsilon))/(1 - \gamma) = 106$ (Ye, 2011), for instance.

results plotted in Fig. 5(c), where we can see that FRP matches the performance of VI across noise levels. It's important to note that this ability to adaptively interpolate between MF and MB behavior, based on the value of $K$, is a unique capability of the FR compared to the SR[3]. The same FRs can be combined using DP to plan for one task or for GPI on the next.

## 4.4 ESCAPE BEHAVIOR

In prey species such as mice, escaping from threats using efficient paths to shelter is critical for survival (Lima & Dill, 1990). Recent work studying the strategies employed by mice when fleeing threatening stimuli in an arena containing a barrier has indicated that, rather than use an explicit cognitive map, mice instead appear to memorize a sequence of subgoals to plan efficient routes to shelter (Shamash et al., 2021). When first threatened, most animals ran along a direct path and into the barrier. Over subsequent identical trials spanning 20 minutes of exploration, threat-stimulus presentation, and escape, mice learned to navigate directly to the edge of the wall before switching direction towards the

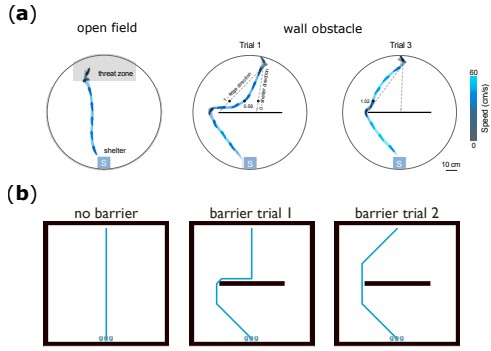

Figure 6: **FRP induces realistic escape behavior.**

shelter (Fig. 6). Follow-up control experiments suggest that mice acquire persistent spatial memories of subgoal locations for efficient escapes. We model this task and demonstrate that FRP induces behavior consistent with these results.

We model the initial escape trial by an agent with a partially learned FR, leading to a suboptimal escape plan leading directly to the barrier. Upon hitting the barrier, sensory input prompts rapid re-planning to navigate around the obstacle. The FR is then updated and the escape plan is recomputed, simulating subsequent periods of exploration during which the mouse presumably memorizes subgoals. We find a similar pattern of behavior to that of mice (Fig. 6(b)). See Appendix A.3 for experimental details.

We do not claim that this is the exact process by which mice are able to efficiently learn escape behavior. Rather, we demonstrate that the FR facilitates behavior that is consistent with our understanding of animal learning in tasks which demand efficient planning. Given the recent evidence in support of SR-like representations in the brain (Stachenfeld et al., 2017; Momennejad et al., 2017), we are optimistic about the possibility of neural encodings of FR-like representations as well. We also re-emphasize that this type of rapid shortest-path planning is not possible with the SR.

## 5 CONCLUSION

In this work, we have introduced the FR, an alternative to the SR which encodes the expected path length between states for a given policy. We explored its basic formal properties, its use as an exploration bonus, and its usefulness for unsupervised representation learning in environments with an ethologically important type of non-Markovian reward structure. We then demonstrated that, unlike the SR, the FR supports a form of efficient planning which induces similar behaviors to those observed in mice escaping from perceived threats. As with any new approach, there are limitations. However, we believe that these limitations represent opportunities for future work. From a theoretical perspective, it will be important to more precisely understand FRP in stochastic environments. For the FF, we have limited understanding of the effect of feature choice on performance, especially in high dimensions. FRP is naturally restricted to discrete state spaces, and it could be interesting to explore approximations or its use in *partially-observable* MDPs with real-valued observations and discrete latents (e.g., Vértes & Sahani, 2019; Du et al., 2019). Further exploration of FRP's connections to hierarchical methods like options would be valuable. Finally, it would be informative to test hypotheses of FR-like representations in the brain. We hope this research direction will inspire advancements on representations that can support efficient behavior in realistic settings.

---

[3]We'd like to stress that this claim of uniqueness is only with respect to the SR. Previous work also explores the use of MF methods to support MB learning (e.g., Pong et al. (2018))

**Reproducibility statement** *Experiments:* All experiments are described in detail in Section 4 in the main text and in Appendix A.3. We have attached code for the tabular experiments (also available at `github.com/tedmoskovitz/first_occupancy`), and in Appendix A.3 we provide a link to the base implementation for APS as well as an explanation of where to find the hyperparameters and modifications made to produce APF. *Theoretical results:* Theoretical results, including assumptions and proofs, are provided in Appendix A.1 and Appendix A.4.

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

## A APPENDIX

### A.1 FR RECURSION

For clarity, we provide the derivation of the recursive form of the FR below:

$$
\begin{aligned}
F^\pi(s, s') &= \mathbb{E}_\pi \left[ \sum_{k=0}^\infty \gamma^k \mathbb{1}(s_{t+k} = s', s' \notin \{s_{t:t+k}\}) \Big| s_t \right] \\
&= \mathbb{E}_\pi \left[ \mathbb{1}(s_t = s', s' \notin \varnothing) + \sum_{k=1}^\infty \gamma^k \mathbb{1}(s_{t+k} = s', s' \notin \{s_{t:t+k}\}) \Big| s_t \right] \\
&= \mathbb{E}_\pi \left[ \mathbb{1}(s_t = s') + \sum_{k=1}^\infty \gamma^k \mathbb{1}(s_{t+k} = s', s_t \neq s', s' \notin \{s_{t+1:t+k}\}) \Big| s_t \right] \\
&= \mathbb{E}_{s_{t+1} \sim p^\pi(\cdot|s)} \left[ \mathbb{1}(s_t = s') + \gamma \mathbb{1}(s_t \neq s') F^\pi(s_{t+1}, s') \Big| s_t \right] \\
&= \mathbb{E}_{s_{t+1} \sim p^\pi(\cdot|s)} \left[ \mathbb{1}(s_t = s') + \gamma (1 - \mathbb{1}(s_t = s')) F^\pi(s_{t+1}, s') \Big| s_t \right]
\end{aligned}
\tag{19}
$$

### A.2 FRP ALGORITHM

We present the full algorithm for FR planning (FRP) below.

---

Algorithm 1: FR Planning (FRP)

---

1: **input:** goal state $s_g$, base policies $\Pi = \{\pi_1, \ldots, \pi_n\}$ and FRs $\{F^{\pi_1}, \ldots, F^{\pi_n}\}$.
2: // initialize discounts-to-goal $\Gamma$
3: $\Gamma_0(s) \leftarrow -\infty \; \forall s \in \mathcal{S}$
4: **for** $s \in \mathcal{S}$ **do**
5: $\quad \Gamma_1(s) \leftarrow \max_{\pi \in \Pi} F^\pi(s, s_g)$
6: $\quad \pi_1^F(s), s_1^F(s) \leftarrow \operatorname{argmax}_{\pi \in \Pi} F^\pi(s, s_g), \; s_g$
7: **end for**
8: // iteratively refine $\Gamma$
9: $k \leftarrow 1$
10: **while** $\exists s \in \mathcal{S}$ such that $\Gamma_k(s) > \Gamma_{k-1}(s)$ **do**
11: $\quad$ **for** $s \in \mathcal{S}$ **do**
12: $\quad\quad \Gamma_{k+1}(s) \leftarrow \max_{\pi \in \Pi, s' \in \mathcal{S}} F^\pi(s, s') \Gamma_k(s')$
13: $\quad\quad \pi_{k+1}^F(s), s_{k+1}^F(s) \leftarrow \operatorname{argmax}_{\pi \in \Pi, s' \in \mathcal{S}} F^\pi(s, s') \Gamma_k(s')$
14: $\quad$ **end for**
15: $\quad k \leftarrow k + 1$
16: **end while**
17: **return** $\pi^F, s^F$

---

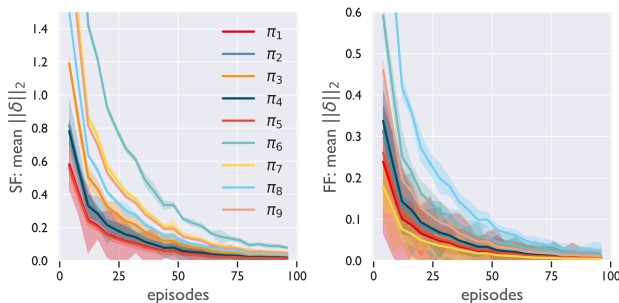

Figure 7: **FF and SF learning curves for continuous MountainCar.** Results averaged over 20 runs. Shading represents one standard deviation.

## A.3 Additional Experimental Details

All experiments except for the robotic reaching experiment were performed on a single 8-core CPU. The robotic reaching experiment was performed using four Nvidia Quadro RTX 5000 GPUs.

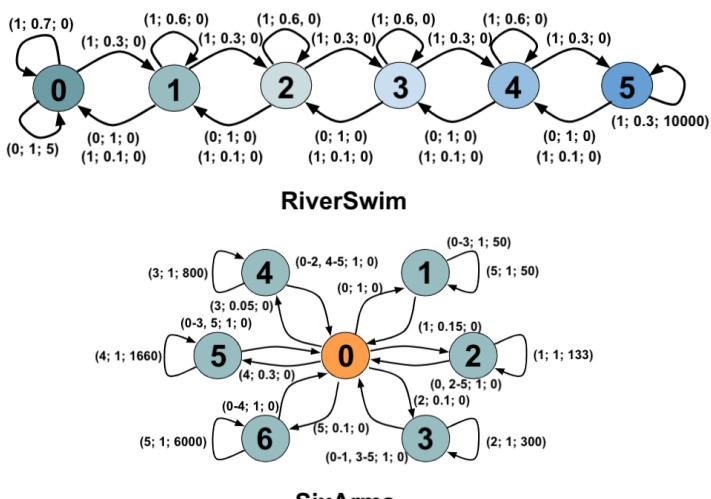

**Figure 8: Tabular environments for exploration.** The tuples marking each transition denote (action id(s); probability; reward). In RiverSwim, the agent starts in either state 1 or state 2 with equal probability, while for SixArms the agent always starts in state 0.

**Tabular exploration** We reuse all hyperparameter settings from Machado et al. (2020) in both the RIVERSWIM and SIXARMS environments, with the only difference being a lower value for $\beta$, the exploration bonus coefficient, for the FR, as the bonuses given by the FR are generally larger. The hyperparameters are $\{\alpha, \eta, \gamma_{\text{SR/FR}}, \beta, \epsilon, \eta\}$, which are the Sarsa learning rate, the SR/FR learning rate, the SR/FR discount factor, the exploration bonus coefficient, and the probability of taking a random action in the $\epsilon$-greedy policy. For RIVERSWIM, these values were $\{0.25, 0.01, 0.95, 100/50, 0.1\}$, respectively, and for SIXARMS they were $\{0.1, 0.01, 0.99, 100/50, 0.01\}$. For the RIVERSWIM-N task, we chose $N = \{6, 12, 24\}$ as default RIVERSWIM has $N = 6$ states, and we chose to successively double the problem size. As $N$ increased, the number of unrewarded central states was multiplied (with the same transition structure), while the endpoints remained the same. It's also worth noting that $\beta$ could be manually adjusted upwards to compensate for the SR bonus' invariance to problem size, though this would require a longer hyperparameter search generally, which we believe is less preferable to a bonus which naturally scales.

Table 2: RIVERSWIM-N results. $\pm$ values denote 1 SE across 100 trials.

| RIVERSWIM-N | SARSA + FR | SARSA + SR | SARSA |
|---|---|---|---|
| $N = 6$ | $1,547,243 \pm 34,050$ | $1,197,075 \pm 36,999$ | $25,075 \pm 1,224$ |
| $N = 12$ | $1,497,937 \pm 29,291$ | $714,797 \pm 34,574$ | $14,590 \pm 3,145$ |
| $N = 24$ | $962,376 \pm 33,325$ | $519,511 \pm 20,580$ | $11,950 \pm 2,643$ |

**Exploration with function approximation** In order to test the usefulness of the FR/FF in a function approximation setting, we use a similar approach to Machado et al. (2020). That is, we train a modified DQN agent (Mnih et al., 2015) using an architecture inspired by Machado et al. (2020) and Oh et al. (2015) (see Fig. 9), such that the base feature representation $\phi(s)$ is an intermediate layer of the network. Like the standard DQN, the architecture outputs an $|\mathcal{A}|$-length vector of predicted $Q$-values for the current state, trained off-policy using minibatches of transition tuples

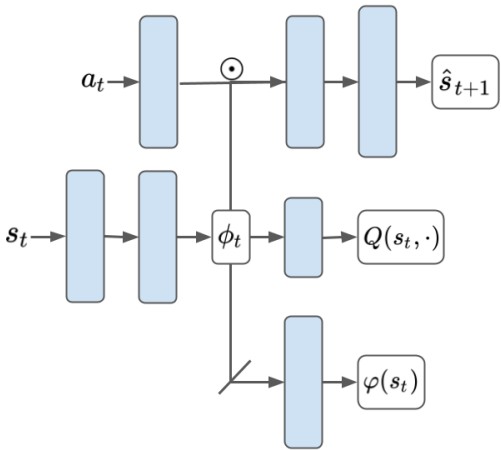

Figure 9: **Network architecture for DQN + FF and DQN + SF**

$\{(s_t^i, a_t^i, r_t^i, s_{t+1}^i)\}_{i=1}^B$ (where $B$ is the minibatch size) to minimize the squared Bellman error

$$\mathcal{L}_Q = \sum_{i=1}^B \| r_t^i + \gamma \max_a Q_-(s_{t+1}^i, a) - Q(s_t^i, a_t^i) \|_2^2 \tag{20}$$

via gradient descent (the subscript $-$ on the target $Q$-values indicates that gradients do not flow through it). Unlike the standard DQN, the network has two additional output heads. The first is a *reconstruction* head which, given the base feature representation of a state $s_t$ and an embedding of the subsequent action $a_t$ produces a prediction of the following state $\hat{s}_{t+1}$. It's trained to minimize the reconstruction loss

$$\mathcal{L}_s = \sum_{i=1}^B \| s_{t+1} - \hat{s}_{t+1} \|_2^2. \tag{21}$$

The final output head of the network is an FF/SF prediction, trained using the squared FF error in the former case:

$$\mathcal{L}_\varphi = \sum_{i=1}^B \| \tilde{\phi}(s_t) + \gamma(1 - \tilde{\phi}(s_t))\varphi_-(s_{t+1}) - \varphi(s_t) \|_2^2, \tag{22}$$

and the squared SF error in the latter

$$\mathcal{L}_\psi = \sum_{i=1}^B \| \phi(s_t) + \gamma \psi_-(s_{t+1}) - \psi(s_t) \|_2^2. \tag{23}$$

For the FF, the features $\phi_t$ are passed through a sigmoid function to compress them in the range $[0, 1]$ and then thresholded at $0.75$. The total loss is then given as a weighted combination

$$\mathcal{L} = w_Q \mathcal{L}_Q + w_s \mathcal{L}_s + w_X \mathcal{L}_X, \tag{24}$$

where $X \in \{\varphi, \psi\}$ and $w_Q, w_s, w_X \in \mathbb{R}$ are fixed weights. As in Machado et al. (2020), gradients from $\mathcal{L}_Q$ and $\mathcal{L}_s$, but not $\mathcal{L}_X$, are permitted to flow through to $\phi$. Thus, the base features are trained to be both reward-predictive and to carry information about the environment transition dynamics. The norm of the FF/SF vector is then used to compute an intrinsic exploration bonus to the task reward in the same manner as in the tabular setting.

To test this model, we chose the DEEPSEA task from the Behavior Suite (`bsuite`; (Osband et al., 2020)) set of benchmark tasks. DEEPSEA is a challenging exploration task set up in an $N \times N$ grid (Fig. 10, top left). At the beginning of each episode, the agent starts at the top left of the grid. Each time step, the agent descends one level, and can choose to move either right or left. The episode ends after $N$ steps, when the agent reaches the bottom level. There is a small negative reward of $-0.01$ if the agent moves right, but if the agent moves right $N$ times in a row, there is a large reward $+1$

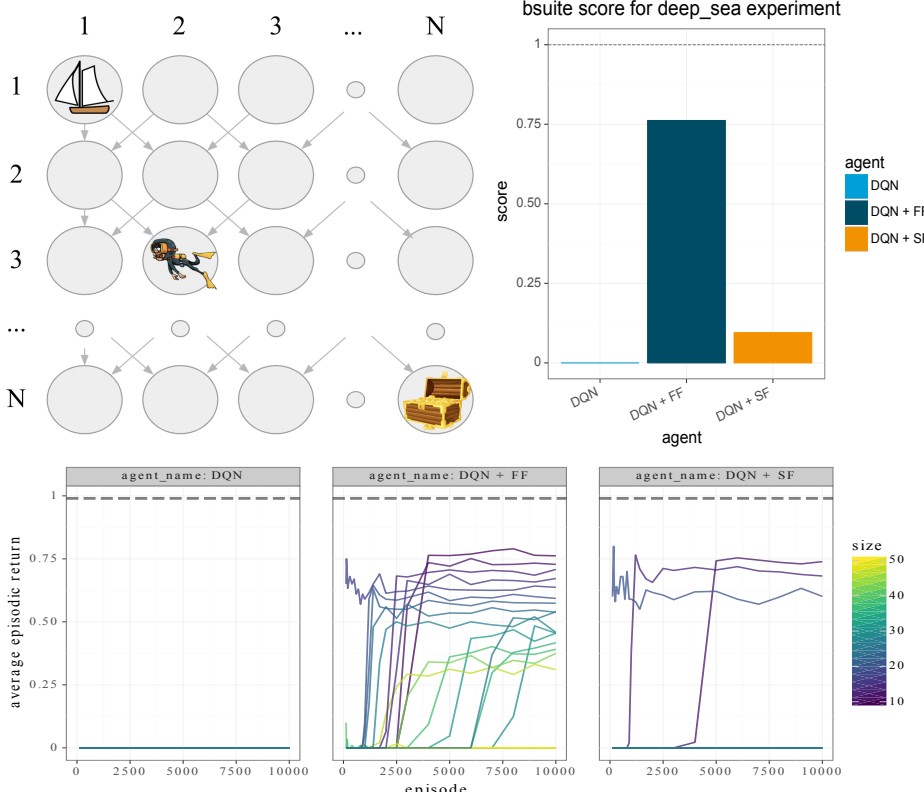

Figure 10: **Exploration with function approximation.** (Top left) Visualization of the DeepSea environment, credit to Osband et al. (2020). (Top right) DQN + FF signficantly outperforms standard DQN and DQN + SF. (Bottom) Different runs across problem sizes.

located at the bottom right of environment—this is the only policy which nets the agent a positive reward. In the `bsuite` framework, the agent is separately trained on increasing problem sizes $N = 5, 6, 7, \ldots, 50$ for 10,000 episodes each, with the final score the proportion of $N$ for which the agent reached an average regret of less than $0.9$ faster than $2^N$ episodes.

We tested the standard DQN, DQN + SF (Machado et al., 2020), and DQN + FF agents on this task, with training hyperparameters described in Table 3. For all models, the network consisted entirely of fully-connected layers, with $\phi(s_t)$ being a 2-layer MLP with 64 units per layer, $Q(s_t, \cdot)$ being a linear function of $\phi(s_t)$ with $|\mathcal{A}| = 2$ units, $\hat{s}_{t+1}$ consisting of a 64-unit layer followed by an $N^2$-unit output layer (the action embedding is 64-dimensional as well), and the SFs/FFs also a 64-dimensional linear layer over $\phi(s_t)$. The agent is trained using $\epsilon$-greedy action selection. Our DQN implementation was coded in JAX (Bradbury et al., 2018) and based off that of Osband et al. (2020). To select the values of $\beta$, $w_s$, and $w_X$ ($w_Q$ was always kept at 1) we performed a sweep over $\beta \in \{0.01, 0.05, 0.1\}$ and $w_X, w_Q \in \{0.001, 0.1, 1, 10, 100, 1000\}$, choosing the best-performing values for each method.

Table 3: Hyperparameter settings for the DEEPSEA experiment

| HYPERPARAMETER | DQN + FF | DQN + SF | DQN |
|---|---:|---:|---:|
| optimizer | Adam (Kingma & Ba, 2017) | Adam | Adam |
| learning rate | 0.001 | 0.001 | 0.001 |
| $\beta$ | 0.05 | 0.01 | – |
| $w_Q, w_s, w_X$ | $(1, 100, 1000)$ | $(1, 0.001, 1000)$ | – |
| $B$ | 32 | 32 | 32 |
| replay buffer size | 10,000 | 10,000 | 10,000 |
| target update period | 4 | 4 | 4 |
| $\gamma$ | 0.99 | 0.99 | 0.99 |
| $\epsilon$ | 0.05 | 0.05 | 0.05 |

Results are presented in Fig. 10. We can see that DQN+FF significantly outperforms the other methods (top right), with the intuition from the tabular experiments—particularly RIVERSWIM-N—carrying over into the function approximation setting. That is, as $N$ increases, the norm of the SF approaches its asymptotic value regardless of the degree of exploration. In contrast, for the FF, the maximum bonus scales with the problem size. This enables the bonus to remain effective in environments with larger state spaces. We hope to investigate this approach and its theoretical properties further in future work.

**MountainCar experiment**  In our version of the task, the feature representations are learned in a rewardless environment, and at test time the reward may be located at any location along the righthand hill. We evaluate the performance of a set of policies $\Pi = \{\pi_i\}$ with a constant magnitude of acceleration and which accelerate in the opposite direction from their current displacement when at rest and in the direction of their current velocity otherwise (see Python code below for details). That is, each $\pi_i$ will swing back and forth along the track with a fixed power coefficient $a_i$. For each possible reward location along the righthand hill, then, the best policy from among this set is the one whose degree of acceleration is such that it arrives at the reward location in the fewest time steps. There is a natural tradeoff–too little acceleration and the cart will not reach the required height. Too much, and time will be wasted traveling too far up the lefthand hill and then reversing momentum back to the right.

*We hypothesized that the FF would be a natural representation for this task, as it would not count the repeated state visits each policy experiences as it swings back and forth to gain momentum.*

We defined a set of policies with acceleration magnitudes $|a_i| = 0.1i$ for $i = 1, \ldots, 9$, and learned both their SFs and FFs via TD learning on an "empty" environment without rewards over the course of 100 episodes, each consisting of 200 time steps, with the SFs using just the simple RBF feature functions without thresholds. Python code for the policy class is shown below.

```python
class FixedPolicy:

def __init__(self, a):
    # set fixed acceleration/power
    self.a = a

def get_action(self, pos, vel):

    if vel == 0:
        # if stopped, accelerate to the opposite end of the environment
        action = -sign(pos) * self.a
    else:
        # otherwise, continue in the current direction of motion
        action = sign(vel) * self.a

    return action
```

We repeated this process for 20 runs, with the plots in Fig. 7 showing the means and standard deviations of the TD learning curves across runs. For the FFs, the thresholds were constant across features at $\theta_d = \theta = 0.7$. Because of the nature of the environment, all of the policies spent a

significant portion of time coasting back and forth between the hills, causing their SFs to accumulate in magnitude each time states were revisited.

Given the learned representations, we then tested them by using them as features for policy evaluation in different tasks, with each task containing a different rewarded/absorbing state. Note that a crucial factor is that the representations were learned in the environment without absorbing states. This is natural, as in the real world reward may arise in the environment anywhere, and we'd like a representation that can be effective for any potential goal location.

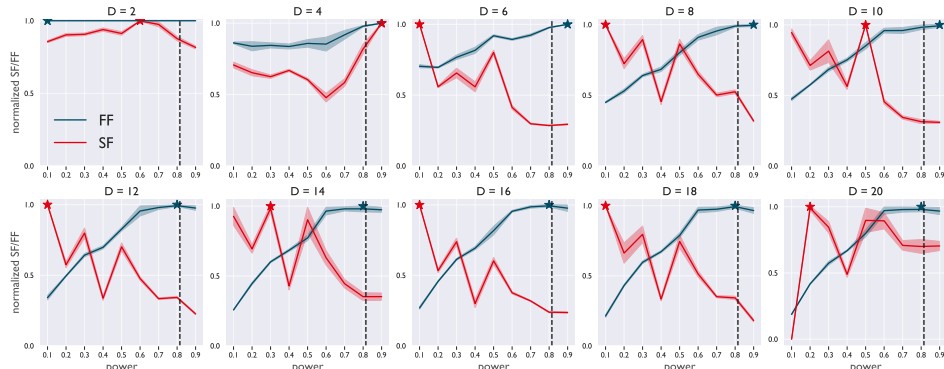

Figure 11: **The FF is robust to feature dimensionality.** FF and SF representation strengths for difference feature dimensionalities between the start and goal locations for an example goal in continuous MountainCar. The vertical dashed line marks the power of the optimal policy. We can see that for all but the coarsest feature representation, the FF is highest for the policy closest to the optimal.

Fig. 11 shows the value of the FF and SF at the start state for different policies with fixed power and for different feature dimensions (number of basis functions) in continuous MountainCar. The results show that the policies for which the FF is highest is closer in power to the optimal policy than for the policies at which the SF is greatest across all but the coarsest feature dimensionalities. This provides an indication of the robustness of the FF to the choice of feature dimensionality.

**Robotic reaching experiment** We used the custom JACO domain as well as the APS base code from Laskin et al. (2021), located at this link: `https://anonymous.4open.science/r/urlb/README.md`. Both the critic and actor networks were parameterized by 3-layer MLPs with ReLU nonlinearities and 1,024 hidden units. Observations were 55-dimensional with 10-dimensional features $\phi(\cdot)$. For all other implementation details, including learning rates, optimizers, etc. see the above link. All hyperparameters and network settings are kept constant from those provided in the linked `.yaml` files. All experiments were repeated for 10 random seeds.

We now describe each training phase. *Pre-training:* Agents were trained for 1M time steps on the rewardless JACO domain by maximizing the intrinsic reward

$$r^{\text{intrinsic}}(s, a, s') = r^{\text{exploit}}(s, a, s') + r^{\text{explore}}(s, a, s')$$

$$= \mathbf{w}^{\mathsf{T}} \phi(s) + \log \left( 1 + \frac{1}{k} \sum_{h^{(j)} \in N_k(\phi(s'))} \|\phi(s') - \phi(s')^{(j)}\|_{n_h}^{n_h} \right), \tag{25}$$

where $w \in \mathbb{R}^D$, $D = 10$ is a random reward vector sampled from a standard Gaussian distribution and the righthand term is a particle-based estimate of the state-based feature entropy, with $N_k(\cdot)$ denoting the $k$ nearest-neighbors (see Liu & Abbeel (2021) for details). In standard APS, this reward is used to train the successor features by constructing the bootstrapped target

$$y^{\text{APS}} = r^{\text{intrinsic}}(s, a, s') + \gamma \mathbf{w}^{\mathsf{T}} \psi(s_{t+1}, a', w), \tag{26}$$

where $a' = \text{argmax}_a \, \mathbf{w}^\mathsf{T}\psi(s', a, w)$. For the FF, we make the following modification:

$$y^{\text{APF}} = y^{\text{APF}-\text{exploit}} + y^{\text{explore}} \tag{27}$$

$$= \underbrace{\mathbf{w}^\mathsf{T}\tilde{\phi}(s)}_{:=r^F(s)} + \gamma\mathbf{w}^\mathsf{T}(\mathbf{1} - \tilde{\phi}(s))V(s_{t+1}) + r^{\text{explore}}(s, a, s') + \gamma V(s_{t+1}) \tag{28}$$

$$= r^F(s) + r^{\text{explore}} + \gamma\left[\mathbf{w}^\mathsf{T}\mathbf{1} - r^F + 1\right]V(s_{t+1}), \tag{29}$$

with $V(s_{t+1}) = \max_{a'} \mathbf{w}^\mathsf{T}\varphi(s_{t+1}, a', w)$, $\tilde{\phi}(\cdot)$ the thresholded base features, such that $\tilde{\phi}_d(s_t) = \mathbb{1}(\phi(s_t) \geq \theta_d, \{\phi_d(s_{t'})\}_{t'=0:t} < \theta_d)$, $\mathbf{1}$ is the $D$-length vector of ones, and $\phi(s_t) \in [0, 1]$. Interestingly, if the features are kept in the range $[0, 1]$, $\phi(\cdot)$ can be thought of encoding a form of "soft" first feature occupancy, rather than the hard threshold given by the indicator function.

The agent is then trained with the off-policy *deep deterministic policy gradient* (DDPG; (Lillicrap et al., 2019)) algorithm, where given a stored replay buffer of transitions $\mathcal{D} = \{(s_t, a_t, r_t, s_{t+1})\}$, the (SF/FF) critic $Q_\omega$ (with $Q$ formed from either the SF or FF and $\omega$ being the parameters) is trained to minimize the squared Bellman loss

$$\mathcal{L}_Q(\omega, \mathcal{D}) = \mathbb{E}_{(s_t, a_t, r_t, s_{t+1}) \sim \mathcal{D}}\left[(y^r - Q_\omega(s_t, a_t))^2\right], \tag{30}$$

where in the pre-training phase $y^r \in \{y^{\text{APS}}, y^{\text{APF}}\}$ (the target parameters are an exponential moving average of the weights—gradients do not flow through them). The deterministic actor $\pi_\theta$ is trained using the derministic policy gradient loss:

$$\mathcal{L}_\pi(\theta, \mathcal{D}) = \mathbb{E}_{s_t \sim \mathcal{D}}\left[Q_\phi(s_t, \pi_\theta(s_t))\right]. \tag{31}$$

*Fine-tuning:* After pre-training, the agent is fine-tuned on the target task, REACHTOPLEFT, where the learning proceeds exactly as in the pre-training phase, but instead of intrinsic reward, the agent is given the task reward—that is, $y^r = r_t^{\text{task}} + \gamma V(s_{t+1})$. We performed this task-specific training for an additional 1M steps.

As additional baselines, we implemented two versions of the dynamic distance learning method of Hartikainen et al. (2020), the original DDLUS and an additional variant DDLUS-G. In standard DDLUS, the goals $g \in \mathcal{S}$ for the pre-training phase are generated according to

$$g^\star \in \underset{g \in \mathcal{D}}{\text{argmax}} \, d^\pi(s_0, g),$$

where $\mathcal{D}$ is a stored set of trajectories. In Hartikainen et al. (2020), this is useful as a mechanism for encouraging the agent to learn skills which move the agent as far as possible from the start state. However, since the object for manipulation in the fine-tuning phase of the JACO task is not typically especially far from the initial point, we also tested DDLUS-G, which samples goals in the same manner as APS and APF, via

$$g \sim \mathcal{N}(0, I),$$

to ensure a more fair comparison. All other hyperparameters match those of Hartikainen et al. (2020), with the exception that the base agent is DDPG, implemented using the same framework as APS and APF, rather than SAC (Haarnoja et al., 2018). For a more detailed discussion of DDL and its relationship to the FR/FF, see Appendix A.8.

In future work, it would be interesting to explore the interaction of the FF with other off-policy algorithms (Haarnoja et al., 2018; Fujimoto et al., 2018; Moskovitz et al., 2021) and whether on-policy learning (e.g., with (Schulman et al., 2017; Kakade, 2002; Williams, 1992; Moskovitz et al., 2020; Hartikainen et al., 2020)) has different effects.

**FourRoom experiments** The FourRoom environment we used was defined on an $11 \times 11$ gridworld in which the agent started in the bottom left corner and moved to a known goal state. The action space was $\mathcal{A} = \{\text{up}, \text{right}, \text{down}, \text{left}\}$ with four base policies each corresponding to one of the basic actions. TD Learning curves for the base policies are depicted in Fig. 12. Once reaching the goal, a new goal was uniformly randomly sampled from the non-walled states. At each time step, the agent received as state input only the index of the next square it would occupy. Each achieved goal netted a reward of $+50$, hitting a wall incurred a penalty of $-1$ and kept the agent in the same

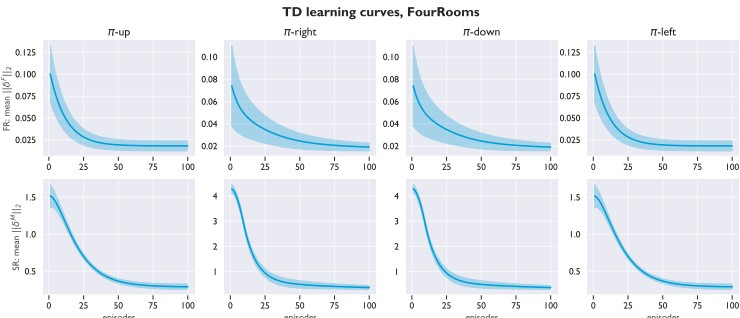

Figure 12: **FOURROOMS learning curves.** FOURROOMS base policies learning curves (average L2 norm of TD errors over 10 runs; shaded area is one standard deviation); top row is for FRs , bottom is for SRs.

place, and every other action resulted in $0$ reward. There were 75 time steps per episode—the agent had to reach as many goals as possible within that limit. The discount factor $\gamma$ was $0.95$, and the FR learning rate was $0.05$. In order to learn accurate FRs for each policy, each policy was run for multiple start states in the environment for 50 episodes prior to training. FRP (for different values of $K$), GPI, and VI were each run for 100 episodes. VI was given the true transition matrix and reward vector in each case. In the stochastic case, for each level of transition noise $\epsilon = 0.0, 0.1, 0.2, \ldots, 1.0$, both VI and FRP were run to convergence ($\approx 180$ iterations for VI, 3 iterations for FRP) and then tested for 100 episodes.

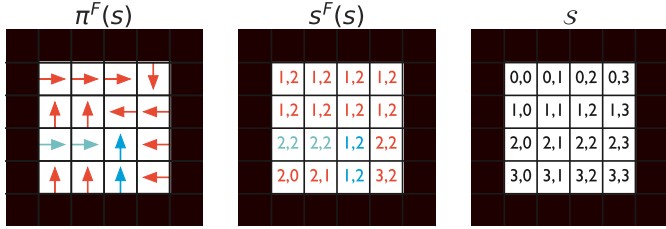

Figure 13: **Implicit planning output.** (Left) The planning policies $\pi^F(s)$ that the agent will elect to follow in each state en route to the goal (see Fig. 3(a)). Arrows denote the action taken by the chosen policy in each state. (Middle) The (row, column) subgoals for each state $s^F(s)$. (Right) The state space $\mathcal{S}$, for reference.

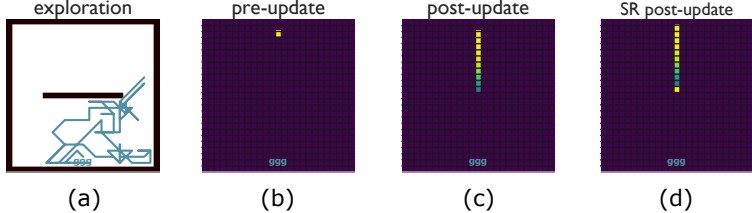

Figure 14: **Exploration and escape** (a) A sample trajectory from the "exploration phase" starting from the shelter. (b) Because the agent starts from the shelter during exploration, the first time it is tested starting from the top of the grid, its FR for the down policy for that state is still at initialization. (c) After updating its FR during testing and further exploration, the FR for the down policy from the start state is accurate, stopping at the barrier. (d) We can see that if we were to use the SR instead, the value in the state above the wall would accumulate when it gets stuck.

**Escape experiments** The escape experiments were modeled as a $25 \times 25$ gridworld with eight available actions,

$$\mathcal{A} = \{\texttt{up}, \texttt{right}, \texttt{down}, \texttt{left}, \texttt{up-right}, \texttt{down-right}, \texttt{down-left}, \texttt{up-left}\} \quad (32)$$

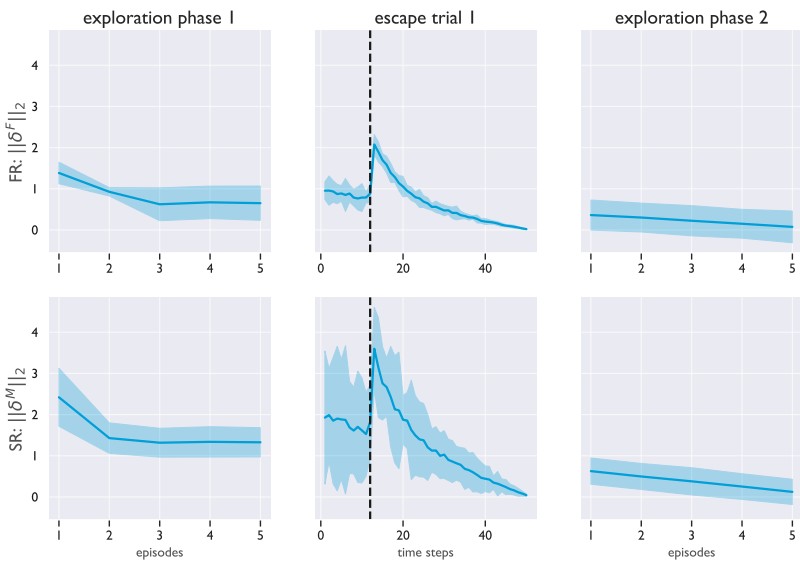

Figure 15: **Escape learning curves.** Learning curves (norms of TD errors) for the first exploration phase, the first escape trial, and the second exploration phase for the "down" policy. The vertical dotted lines in the escape trial mark the time step at which the agent encounters the barrier. This causes a temporary jump in the TD errors, as representation learning did not reflect the wall at this point. The top row consists of FR results and the bottom row is from SRs, averaged over 10 runs. The shading represents one standard deviation.

and a barrier one-half the width of the grid optionally present in the center of the room. The discount factor $\gamma$ was $0.99$, and the FR learning rate was $0.05$. In test settings, the agent started in the top central state of the grid, with a single goal state directly opposite. At each time step, the agent receives only a number corresponding to the index of its current state as input. The base policy set $\Pi$ consisted of eight policies, one for each action in $\mathcal{A}$. Escape trials had a maximum of 50 steps, with termination if the agent reached the goal state. In (Shamash et al., 2021), mice were allowed to explore the room starting from the shelter location. Accordingly, during "exploration phases," the agent started in the goal state and randomly selected a base policy at each time step, after which it updated the corresponding FR. Each exploration phase consisted of 5 episodes—a sample trajectory is shown in Fig. 14(a). After the first exploration phase, the agent started from the canonical start state and ran FRP to reach the goal state. Because most of its experience was in the lower half of the grid, the FRs for the upper half were incomplete (Fig. 14(b)), and we hypothesized that in this case, the mouse should either i) default to a policy which would take it to the shelter in the area of the room which it knew well (the down policy) or ii) default to a policy which would simply take it away from the threat (again the down policy). During the first escape trial, the agent selects the down policy repeatedly, continuing to update its FRs during the testing phase. Upon reaching the wall and getting stuck, the FR for the down policy is eventually updated enough that re-planning with FRP produces a path around the barrier. After updating its FR during the first escape trial and during another exploration period, the FRs for the upper half of the grid are more accurate (Fig. 14(c)) and running FRP again from the start state produces a faster path around the barrier on the second escape trial. TD learning curves for the experiment (repeated with the SR for completeness) are plotted in Fig. 15.

For completeness, we repeated this experiment with the SR. In this case, the planning algorithm is ill-defined for $K > 0$, so we default to GPI ($K = 0$). As expected, without the barrier, the down policy is selected and the goal is reached (Fig. 16,left). However, when there is a barrier, while the SR updates when the agent hits it (Fig. 12), since there is no single policy that can reach the shelter, GPI fails to find a path around the barrier (Fig. 16,middle,right).

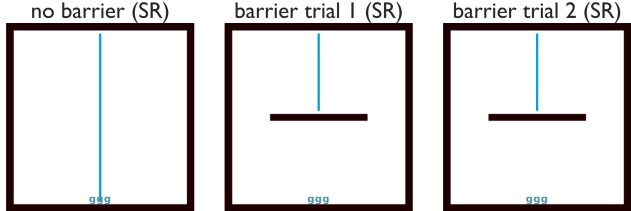

Figure 16: **An SR cannot effectively escape under the same conditions as an FR agent.**

## A.4 ADDITIONAL PROOFS

Below are the proofs for Proposition 3.1 and Proposition 3.2, which are restated below.

**Proposition 3.1** (Contraction). *Let $\mathcal{G}^\pi$ be the operator as defined in Definition 3.2 for some stationary policy $\pi$. Then for any two matrices $F, F' \in \mathbb{R}^{|\mathcal{S}| \times |\mathcal{S}|}$,*

$$|\mathcal{G}^\pi F(s, s') - \mathcal{G}^\pi F'(s, s')| \leq \gamma |F(s, s') - F'(s, s')|, \tag{33}$$

*with the difference equal to zero for $s = s'$.*

*Proof.* For $s \neq s'$ we have

$$|(\mathcal{G}^\pi - \mathcal{G}^\pi F')_{s,s'}| = \gamma |(P^\pi F - P^\pi F')_{s,s'}| = \gamma |P^\pi (F - F')_{s,s'}| \leq \gamma |(F - F')_{s,s'}|,$$

where we use the notation $X_{s,s'}$ to mean $X(s, s')$, and the inequality is due to the fact that every element of $P^\pi(F - F')$ is a convex average of $F - F'$. For $s = s'$, we trivially have $|(\mathcal{G}^\pi F - \mathcal{G}^\pi F')_{s,s}| = |1 - 1| = 0$. $\square$

**Proposition A.1** (Convergence). *Under the conditions assumed above, set $F^{(0)} = I_{|\mathcal{S}|}$. For $k = 0, 1, \ldots$, suppose $F^{(k+1)} = \mathcal{G}^\pi F^{(k)}$. Then*

$$|F^{(k)}(s, s') - F^\pi(s, s')| < \gamma^k \tag{34}$$

*for $s \neq s'$ with the difference for $s = s'$ equal to zero $\forall k$.*

*Proof.* We have, for $s \neq s'$ and using the notation $X_{s,s'} = X(s, s')$ for a matrix $X$,

$$
\begin{aligned}
|(F^{(k)} - F^\pi)_{s,s'}| &= |(\mathcal{G}^k F^{(0)} - \mathcal{G}^k F^\pi)_{s,s'}| \\
&\leq \gamma^k |(F^{(0)} - F^\pi)_{s,s'}| \quad \text{(Proposition 3.1)} \\
&= \gamma^k F^\pi(s, s') < \gamma^k \quad (F^\pi(s, s') \in [0, 1)).
\end{aligned} \tag{35}
$$

For $s = s'$, $|(F^{(k)} - F^\pi)_{s,s}| = |1 - 1| = 0 \; \forall k$. $\square$

Below is the proof of Proposition 4.1, which is restated below.

**Proposition 4.1** (Planning optimality). *Consider a deterministic, finite MDP with a single goal state $s_g$, and a policy set $\Pi$ composed of policies $\pi : \mathcal{S} \to \mathcal{A}$. We make the following coverage assumption, there exists some sequence of policies that reaches $s_g$ from a given start state $s_0$. Under these conditions, Alg. 1 converges such that $\Gamma(s_0) = \gamma^{L_\pi^\star}$, where $L_\pi^\star$ is the shortest path length from $s_0$ to $s_g$ using $\pi \in \Pi$.*

*Proof.* Since the MDP is deterministic, we use a deterministic transition function $\rho : \mathcal{S} \times \mathcal{A} \to \mathcal{S}$. We proceed by induction on $L_\pi^\star$.

Base case: $L_\pi^\star = 1$

If $L_\pi^\star = 1$, $s_0$ must be one step from $s_g$. The coverage assumption guarantees that $\exists \pi \in \Pi$ such that $\rho(s_0, \pi(s_0)) = s_g$. Note also that when both the MDP and policies in $\Pi$ are deterministic, $F^\pi(s, s') = \gamma^{L_\pi}$, where $L_\pi$ is the number of steps from $s$ to $s'$ under $\pi$, and we use the abuse of notation $L_\pi = \infty$ if $\pi$ does not reach $s'$ from $s$.

Then following Alg. 1,

$$\Gamma_1(s_0) = \max_{\pi \in \Pi} F^\pi(s_0, s_g) = \gamma \quad \text{(guaranteed by coverage of } \Pi\text{)}$$

$$\Gamma_1(s_g) = \max_{\pi \in \Pi} F^\pi(s_g, s_g) = 1 \quad \text{(by definition of } F^\pi\text{)}.$$

Moreover,

$$\begin{aligned}
\Gamma_2(s_0) &= \max_{\pi \in \Pi, s' \in \mathcal{S}} F^\pi(s_0, s')\Gamma_1(s') \\
&= \max_{\pi \in \Pi}\{F^\pi(s_0, s_0)\Gamma_1(s_0), F^\pi(s_0, s_g)\Gamma_1(s_g)\} \\
&= \max\{1 \cdot \gamma, \gamma \cdot 1\} \\
&= \gamma.
\end{aligned}$$

Then $\Gamma_2(s) = \Gamma_1(s) \ \forall s$ and Alg. 1 terminates. Thus, $\Gamma(s_0) = \gamma = \gamma^{L_\pi^\star}$ and the base case holds.

Induction step: Assume Proposition 4.1 holds for $L_\pi^\star = L$

Given the induction assumption, we now need to show that Proposition 4.1 holds for $L_\pi^\star = L + 1$. By the induction and coverage assumptions, there must exist at least one state within one step of $s_g$ that the agent can reach in $L$ steps, such that the discount for this state or states is $\gamma^L$. Moreover, the coverage assumption guarantees that $\exists \pi \in \Pi$ such that for at least one such state $s_L$, $\rho(s_L, \pi(s_L)) = s_g$.

Then this problem reduces to the base case—that is, Alg. 1 will select the policy $\pi \in \Pi$ that transitions directly from $s_L$ to $s_g$—and the proof is complete. $\qquad\square$

### A.5 EXPLICIT PLANNING

Below we describe a procedure for constructing an explicit plan using $\pi^F$ and $s^F$.

---

**Algorithm 2:** `ConstructPlan`

---

1: **input:** goal state $s_g$, planning policy $\pi^F$, subgoals $s^F$
2: $\Lambda \leftarrow []$          $\triangleright$ init. plan
3: $s \leftarrow s_0$          $\triangleright$ begin at start state
4: **while** $s \neq s_g$ **do**
5:     $\Lambda.\text{append}((\pi^F(s), s^F(s)))$          $\triangleright$ add policy-subgoal pair for current state to plan
6:     $s \leftarrow s^F(s)$
7: **end while**
8: **return** $\Lambda$

---

### A.6 FRP WITH MULITPLE GOALS

Here we consider the application of FRP to environments with multiple goals $\{g_1, \ldots, g_n\}$. To find the shortest path between them given the base policy set $\Pi$, we first run FRP for each possible goal state in $\{g_i\}$, yielding an expected discount matrix $\Gamma^\Pi \in [0, 1]^{|\mathcal{S}| \times n}$, such that $\Gamma^\Pi(s, g_i)$ is the expected discount of the shortest path from state $s$ to goal $g_i$. We denote by $g_\sigma = [s_0, \sigma(g_1), \sigma(g_2), \ldots, \sigma(g_n)]$ a specific ordering of the goals in $\{g_i\}$ starting in $s_0$. The expected discount of a sequence of goals is then

$$\Xi(g_\sigma) = \prod_{i=1}^n \Gamma^\Pi(g_\sigma(i-1), g_\sigma(i)), \tag{36}$$

with the optimal goal ordering $g_{\sigma^*}$ given by

$$g_{\sigma^*} = \operatorname*{argmax}_{g_\sigma \in G} \Xi(g_\sigma), \tag{37}$$

where $G$ is the set of all possible permutations of $\{g_i\}$, of size $n!$. This is related to a form of the travelling salesman problem, and we refer the reader to Zahavy et al. (2019) for a formal investigation of the use of local policies to construct shortest paths. Fortunately, in most settings we don't expect the number of goals $n$ to be particularly large.

Table 4: Overview of basic points of comparison between the FR/FF, SR/SF (Dayan, 1993; Barreto et al., 2017b), DDL (Hartikainen et al., 2020), TDMs (Pong et al., 2018), and DG (Kaelbling, 1993).

|  | FR/FF | SR/SF | DDL | TDM | DG |
|---|---|---|---|---|---|
| On- v. Off-policy | Both | Both | On-policy | Off-policy | Off-policy |
| Eval. v. Control | Eval. | Eval. | Eval. | Control | Control |
| Finite v. Inf. Horizon | Both | Both | Finite Horizon | Finite Horizon | Finite Horizon |
| State representation? | Yes | Yes | No | No | No |

### A.7 CONNECTIONS TO OPTIONS

The options framework (Sutton et al., 1999) is a method for temporal abstraction in RL, wherein an option $\omega$ is defined as a tuple $(\pi_\omega, \tau_\omega)$, where $\pi_\omega$ is a policy and $\tau_\omega \in \Delta(\mathcal{S})$ is a state-dependent termination distribution (or function, if deterministic). Executing an option at time $t$ entails sampling an action $a_t \sim \pi_\omega(\cdot|s_t)$ and ceasing execution of $\pi_\omega$ at time $t+1$ with probability $\tau_\omega(s_{t+1})$. The use of options enlarges an MDP's action space, whereby a higher-level policy selects among basic, low-level actions and options.

Options are connected to FRP (Alg. 1) in that by outputting a set of policies and associated subgoals $\{(\pi^F, s^F)\}$, FRP effectively converts each base policy to an option with a deterministic termination function, i.e., the agent will follow $\pi^F$ until terminating at $s^F$. One of the key difficulties in the options literature is how to learn the best options to add to the available action set. For the class of problems considered in this paper, FRP then provides a framework for generating *optimal* (in the sense of finding the fastest path to a goal) options from a set of standard policies, subject to the fulfillment of the coverage assumption. Importantly, the associated FRs (which can be learned via simple TD updating) can be reused across tasks, so that FRP can re-derive optimal options for a new goal.

FRP can also be seen as related to the work of Silver & Ciosek (2012), which demonstrates that value iteration performed on top of a set of task-specific options converges more quickly than value iteration performed on the default state space of the MDP. One critical difference to note is that the FR/FRP is transferable to any MDP with shared transition dynamics. While value iteration on a set of options for a given MDP is more efficient than value iteration performed directly on the underlying MDP, this process must be repeated every time the reward function changes. However, the FR enables an implicit representation of the transition dynamics to be cached and reused.

### A.8 FURTHER CONNECTIONS TO RELATED WORK

We now describe the connection between the FR/FF and several related approaches in the literature. Table 4 summarizes a high-level view of these connections.

**The Dynamic Distance Function** The *dynamic distance function* (DDF; (Hartikainen et al., 2020)) is defined as

$$d^\pi(s, s') = \mathbb{E}_\pi \left[ \sum_{k=0}^{j-1} \gamma^k c(s_{t+k}, s_{t+k+1}) \Big| s_t = s, s_j = s' \right], \tag{38}$$

where $c : \mathcal{S} \times \mathcal{S} \to \mathbb{R}$ is a local cost function. In practice, $c(s_{t+k}, s_{t+k+1}) = \gamma = 1$, giving

$$d^\pi(s, s') = \mathbb{E}_\pi \left[ \sum_{k=0}^{j-1} 1 \Big| s_t = s, s_j = s' \right]. \tag{39}$$

n practice, $d^\pi(s, s')$ is parameterized via a neural network with parameters $\psi$ trained on-policy from full trajectories $\tau$ using the loss

$$\mathcal{L}(\psi) = \frac{1}{2} \mathbb{E}_{\tau \sim \mathcal{D}, i \sim [0,T], j \sim [i,T]} (d_\psi^\pi(s_i, s_j) - (j - i))^2, \tag{40}$$

where $\mathcal{D}$ is a buffer of stored trajectories. On a downstream navigation task, the agent policy is trained to minimize

$$\mathcal{L}_\pi(\phi) = \mathbb{E}_\pi \left[ \sum_{t=0}^\infty \gamma^t d_\psi^\pi(s_t, g), \right], \tag{41}$$

where $g \in \mathcal{S}$ is a goal state and $\phi$ are the policy parameters.

There are several important differences between the DDF and the FR, both in theory and practical application.

This form of the DDF, as discussed by Hartikainen et al. (2020), is also naturally restricted to on-policy learning from full trajectories, and cannot be updated off-policy and/or via one-step temporal difference learning. A natural additional consequence is that it is only applicable to finite-horizon MDPs. This confinement to finite horizons is significant because it leads to the conditioning problem described by Hartikainen et al. (2020). This problem occurs because the DDF is conditioned on the policy successfully reaching the goal state—when this doesn't happen, it can lead to significant value estimation errors.

A second difference is that the policy is trained, via Eq. (41), to minimize the *cumulative* discounted distance to the goal rather than via greedy distance minimization.

Another difference is that Eqs. (38) and (39) don't explicitly require time step $j$ to be the first time the agent enters $s'$, although this may have been the authors' intention. In fact, this definition is more closely related to the SR than the FR (with equality in the infinite horizon setting), but the implementation of the DDF in practice is more closely related to the FR. It's also important to note that when $\gamma = 1$, the number of steps $k$ appears linearly within the expectation rather than exponentially (this is important, as in general $\gamma^{\mathbb{E}[k]} \neq \mathbb{E}[\gamma^k]$.

Another significant difference, then, is that the DDF as presented cannot be in a meaningful sense be considered a state *representation*, but rather a function mapping pairs of states to expected distances. That is, $d^\pi(s, \cdot)$ has no meaningful semantics when implemented as a mapping $d^\pi : \mathcal{S} \times \mathcal{S} \to \mathbb{R}$ and trained using Eq. (40). In practice, it is used to support policy optimization, rather than evaluation. In contrast, the FF representation, like the SF representation, maps a single state to a fixed length vector-valued encoding, $\varphi^\pi : \mathcal{S} \to \mathbb{R}^d$. This difference has significant implications for the applications of the DDF. In particular, it is unclear how the DDF could be used as an exploration bonus in the manner of the FF/SF, and the scalar value would make it challenging to implement the type of parallel updates useful for efficient planning.

**Dynamic Goal Learning**    Another related approach is that of *dynamic goal* (DG) learning (Kaelbling, 1993), a method for optimal control related to $Q$-learning. The optimal DG function $G^\star : \mathcal{S} \times \mathcal{A} \times \mathcal{S} \to \mathbb{R}$ is defined recursively for a goal state $g \in \mathcal{S}$ as

$$G^\star(s, a, g) = 1 + \mathbb{E}_{s' \sim P(\cdot|s,a)} \left[ \min_{a' \in \mathcal{A}} G^\star(s', a', g) \right], \tag{42}$$

where $G^\star(g, a, g) \coloneqq 0$. There are several important differences between this approach and the FR. First, as mentioned above, DG learning is a method for optimal control, rather than policy *evaluation*. That is, $D^\star(s, a, g)$ converges to the expected number of steps for the optimal policy for reaching $g$ starting from $s$. It cannot be reused for policy evaluation for a policy $\pi \neq \pi^\star$, and also implicitly assumes that $g$ is reachable in finite time. DG learning is thus susceptible to the same conditioning problem discussion above for the DDF, and is only studied by Kaelbling (1993) in a gridworld environment. A second difference is that the DG function scales linearly with $k$, the number of steps for the optimal policy to reach $g$, while the FR scales exponentially at a rate of $\gamma$. This is important, as we can note (with a slight abuse of notation) in general that $\gamma^{\mathbb{E}_\pi[k]} \neq \mathbb{E}_\pi \left[ \gamma^k \right]$—it is nontrivial to recover the FR for the optimal policy from the DG function (and vice-versa).

**Temporal Difference Models**    *Temporal difference models* (TDMs; (Pong et al., 2018)) are another related approach. TDMs are an optimal control method motivated by the observation that goal-conditioned $Q$-functions can be used to construct an implicit dynamics model when the discount factor $\gamma = 0$. For $\gamma > 0$, the authors introduce a horizon variable $\tau$ to interpolate between model-based and model-free learning. Like the FR, this formulation allows agent to consider multiple

goals in parallel. As in DG learning, and unlike the FR, TDMs represent a method for optimal control, not evaluation. The emphasis is on efficiently improving a policy, rather than learning a state representation which can be leveraged for multiple uses (e.g., policy evaluation or as an exploration bonus). Another difference from the FR is that, like the DDF and DG learning, TDMs rely on a finite-horizon setting.

## A.9 FR VS. SR VISUALIZATION

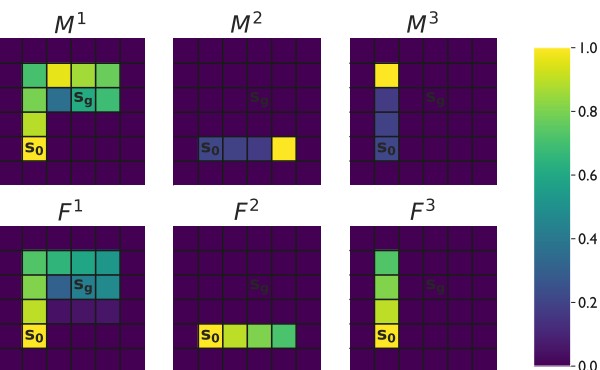

Figure 17: **SR vs. FR visualization** The SRs and FRs from the start state for the policies in Fig. Fig. 3. For the SRs (Fig. 17, top row), we can see that states that are revisited (or in which the policy simply stays) are more highly weighted, while for the FRs (Fig. 17, bottom row), the magnitude of $F(s_0, s')$ is higher for states $s'$ that are closer along the path taken by the policy.

