# OpenReview forum: "A First-Occupancy Representation for Reinforcement Learning"
_ICLR.cc/2022/Conference — ICLR 2022 Poster_

### Official Review · Reviewer_epv4 · 2021-10-26

**Correctness:** 4
**Technical Novelty And Significance:** 4
**Empirical Novelty And Significance:** 2
**Recommendation:** 8
**Confidence:** 4

**Main Review:**

## Short Version
Overall, I think this is a very nice paper that presents a neat little idea with a lot of potential impact in the community. Successor features have resurfaced in the RL community in the past few years, and the proposed "FF" features improve upon them in a very specific, but evident way. My biggest concern is the somewhat limited empirical evaluations and broader comparisons that bring out the strengths of the method.
[Full Disclosure: I have not checked the accuracy of the theoretical claims thoroughly; they seem appropriate on a quick glance.]

---
## Strengths
1. The writing is very succinct and the authors make a great case of comparing against the SR in every way possible (and do so fairly).
2. The paper is well-written in terms of reproducibility and the authors make plenty of information available about the experiments and algorithm to reproduce the results. The theoretical results are also discussed elaborately in the appendix.
3. The suite of capabilities considered, and demonstrated, is very complete and makes a good case for "FR > SR" -- exploration bonus, unsupervised pretraining and planning. Particularly, the section on planning with FR makes a good case for their strengths and could be made more prominent in the contributions/introduction, in my opinion. The paper makes a great case for the _completeness_ of a first-occupancy representation.
4. The parallels to animal behavior, although a bit far-fetched to me, are very nice inclusions in the article. I appreciate the candor of the authors in presenting the facts ("We do not claim that this is the exact process by which mice are able to efficiently learn escape behavior. Rather, we demonstrate that the FR facilitates behavior that is consistent with our under- standing of animal learning") and not overfitting their findings to tall claims about understanding the animal brain. The findings are interesting in their own right and add value to the paper.


---
## Weaknesses
1. While the paper does a great job at comparing against the vanilla successor representation and shows (quite clearly) that FR > SR in tabular domains, my biggest concern is that the results don't go too far beyond that. In most realistic domains, the infinite state-space version (FF) would be more relevant and there is almost no empirical analysis of how the method performs in more complex domains/tasks, beyond Section 4.2. The authors also do not compare against a broader suite of methods in the recent years that use successor features to do more interesting control and planning problems; a more thorough empirical evaluation will greatly increase the impact of the work and serve as a strong baseline for future works to present results against. As a suggestion, I would recommend [a recent paper by Janner et al.](https://arxiv.org/abs/2010.14496) which conducts thorough empirical evaluation of a model similar to the SR, and can provide some pointers towards more tasks to consider for empirical analysis.
2. While motivating the need for a _better_ representation, the authors frequently make claims about how the problem of non-Markovian rewards is widespread and _natural_, e.g. "Such natural problems emphasize the importance of FR...". However, the text fails to provide a concrete example for the reader to conceptualize the claims and contributions of the paper. As a suggestion on the presentation, I would urge the authors to present a concrete example that motivates the need for the first-occupancy early on, or a toy problem that can isolate the issue, for clarity.

**Summary Of The Paper:**

The paper proposes an alternative to the successor representation (SR) -- the first-occupancy representation (FR) -- which represents the expected time to first visitation of a state. The paper motivates the representation to be applicable in environments with non-Markovian rewards and show that the proposed FR can handle such cases. The authors present a well-executed comparison to SR and demonstrate improved exploration in MF settings and the ability to plan in a model-based setting. The authors also provide some theoretical analysis of the representation, most notably a convergence result on the Bellman operator for the FR, and planning optimality in a model-based setting.

**Summary Of The Review:**

The paper presents a simple idea which is supported by ample theoretical analysis and (some) experiments. Comparison to prior work, while not exhaustive, is convincing. I vote to accept the paper but strongly urge the authors to improve the empirical analysis for the overall impact of the paper.

---

> ### Author Response · Authors · 2021-11-16
> **Authors' Response**
>
>    Thank you very much for your detailed comments and suggestions--we're glad that you enjoyed the paper! Please find a response to your concerns below.
>
>    1. *Empirical evaluation*: We have added two new experimental results to the revision--an additional pair of baselines on the robotic reaching task (Figure 4) and an evaluation of the FF vs. the SF on the difficult exploration task DeepSea [1] using function approximation with a DQN-style architecture (Appendix A.3). We hope to explore further high-dimensional tasks in the function approximation setting such those in Janner et al. in future work--thank you for the reference!
>    2. *Natural problems and a toy example*: We agree that motivating the FR is an important part of the paper, and we apologize for not doing so in a sufficiently clear manner. Some basic examples of natural non-Markovian rewards are reaching a goal state as quickly as possible or an animal foraging for food, where the food (reward) disappears from a state after consumption. As an intuitive example, when planning a route to a restaurant to meet a friend, the FR is largest for a route/policy that arrives at the front door the fastest, whereas the SR is largest for the route which walks past the front door the highest number of times. This can be seen as a real-world instance of the toy problem we depict in Figure 1 (described in Section 3), and we will certainly add more detailed motivation to the paper. We'd also like to emphasize that our position isn't that the SR should be dropped entirely as a representation in favor of the FR, but rather that there are certain problem types for which the FR is a more natural representation. We are big fans of the SR/SFs, in fact!
>
> Thank you once again for your assessment and helpful comments, and we hope that we've addressed your concerns.
>
> [1] https://openreview.net/forum?id=rygf-kSYwH

---

> > ### Comment · Reviewer_epv4 · 2021-11-16
> > **Thanks!**
> >
> > Thank you for the discussion. I am in favor of acceptance and stand by my original recommendation.

---

### Official Review · Reviewer_SXgZ · 2021-10-28

**Correctness:** 4
**Technical Novelty And Significance:** 4
**Empirical Novelty And Significance:** 4
**Recommendation:** 8
**Confidence:** 4

**Main Review:**

The proposed representation (FR) seems to have clear advantages in comparison to approaches that convert MDPs with non-Markovian reward structure to such with Markovian one. The use of the first-occupancy values as an exploration bonus results in much more efficient exploration. The authors also demonstrate that FR can be used effectively for unsupervised pre-training RL with non-Markovian rewards, where the successor representations tends to produce incorrect estimates of the value function. The use of FR for planning is also demonstrated on a high-dimensional continuous control problem involving a robotic arm. All of these illustrate well the advantages of the proposed representation, and overall, I find this representation highly original.

One limitation of this general approach is probably the assumption that a fixed, and relatively small number of policies will be provided in advance, and the solution of the sequential decision problem can be expressed by switching between these policies. It is not clear if and under what conditions this assumption is justified. Could the (sub-)optimality of this approach be analyzed theoretically or empirically, by computing the true optimal policy and comparing with its performance? Section 3 talks about "Policy evaluation and improvement with the FR", and although the use of FR for fast policy evaluation is explained very well, I do not see any discussion about policy improvement there.

A minor typo, on page 4, last paragraph: "automoton" -> "automaton".



**Summary Of The Paper:**

The paper proposes a novel representation of the dynamics of an environment that is independent of the reward structure encoding a particular sequential decision task in it, and thus can be reused to speed up the computation of policies for multiple tasks in this environment. Unlike successor representations, which compute the likelihood of occupying a particular state at any time in the future, when starting from a given initial state, the proposed first-occupancy representation encodes the likelihood of reaching a particular state for the first time. Because of this, the novel representation effectively represents the expected path length between all pairs of states when following a particular policy. The benefits of the proposed representation are illustrated in several decision problems with non-Markovian reward structure. It is also argued that animals might use similar representations of their environment to effectively find escape routes in a short amount of time.

**Summary Of The Review:**

I find the proposed novel representation highly original, and ICLR is the right venue for its publication. It is a clear advance of the state of the art in computing intermediate representations that cache the system dynamics in a format that is more suitable for policy evaluation than the basic transfer function of the MDP. Its advantages over successor representations for the case of non-Markovian rewards are clearly explained and supported by empirical evaluations. For these reasons, I recommend acceptance. It is also apparent that the proposed approach has some limitations, and the authors readily acknowledge this.

---

> ### Author Response · Authors · 2021-11-16
> **Authors' Response**
>
> Thank you very much for your helpful comments and suggestions--we are glad you liked the paper! Please find a response below.
>
>    1. *Limitations of this general approach*: Indeed, this is an interesting question. In the case of finite MDPs, if $\Pi$ contains policies which each take one action within the action space, then FRP will converge to $\pi^\star$, for example. A more difficult question, which we consider outside the scope of our work, is finding the necessary conditions for continuous MDPs.  In continuous state spaces, an important quantity to consider is the state-wise span of the constituent policies. In most deep RL frameworks for continuous control, which frequently sample actions from a Gaussian distribution in action space, a given (stochastic) policy could move in any direction. These conditions are generally sufficient to ensure that a (near-)optimal policy can be composed. We also refer to the references Barreto et al. (2017a,b), Barreto et al. (2018), and Barreto et al. (2020) which also use state representations as the basis for a compositional approach to multitask learning, and in particular Zahavy et al. (2021), which explicitly addresses the question of which policies should comprise the set $\Pi$.
>
>    - *Policy evaluation and improvement*: Yes, thank you for pointing this out--a reference to policy improvement was cut during the editing process, and we have added it back in the revision. (Policy improvement with respect to $Q_{r^F}^\pi$ functions in the same manner as standard settings.)
>    - *Typo*: Thank you! We've fixed it.
>
>    Thank you once again!

---

> > ### Comment · Reviewer_SXgZ · 2021-11-24
> > **Suggestions addressed well**
> >
> > Thank you for addressing my comments and suggestions in a very satisfactory manner. I maintain my rating and remain in favor of acceptance.

---

### Official Review · Reviewer_aieJ · 2021-11-02

**Correctness:** 3
**Technical Novelty And Significance:** 2
**Empirical Novelty And Significance:** 2
**Recommendation:** 5
**Confidence:** 4

**Main Review:**

This paper proposes a general-purpose modification of the successor representation that has a number of nice properties: in particular, it can be used for shortest path search and can be stitched together to form a multi-step plan in the spirit of options. The paper also shows demonstrations of its usefulness in unsupervised RL and exploration. The evidence for each individual application is somewhat preliminary, which I think is fine for a paper that intends primarily to introduce a new prediction problem and outline potential applications. The most compelling idea to me is the planning (FRP) procedure, which naturally fits into other DP-style methods and multi-step planning frameworks.

The main issue I can find is that the proposed first-occupancy representation might not be as new as the paper describes. As early as [Kaelbling 1993](http://citeseerx.ist.psu.edu/viewdoc/summary?doi=10.1.1.51.3077) there was the so-called "dynamic goal learning" (DG) algorithm, which uses TD to predict the minimum number of timesteps required to reach a goal. The FR is in the space of discounts and DG in the space of timesteps, but at least for deterministic policies and dynamics it seems like there would be a straightforward conversion between the two ($FR = \gamma^{DG}$?).

For a more modern take on the same problem, [dynamical distance learning (DDL)](https://arxiv.org/abs/1907.08225) also estimates the expected number of timesteps required to reach a goal state, and [temporal difference models (TDMs)](https://arxiv.org/abs/1802.09081) estimate how close a policy will come to a goal state in a fixed number of timesteps.

It is not necessarily an issue that there are papers from the last few years investigating a similar line of inquiry; the paper is still a thorough description of this idea and has enough in it that it is not subsumed by these prior works. But since the space has already been scoped out a bit, the framing of this paper as a first introduction of the idea might not be the right one, and raises the bar slightly in terms of experimental evaluation. It would be valuable to compare it to the related approaches that have already been scaled to higher dimensions (like DDL and TDMs).

**Minor**
1. "to take advantage this shared" --> "advantage of"
2. In Equation 8, should the condition read $s_t = s$?


**Summary Of The Paper:**

This paper presents the first-occupancy representation, a sort of non-Markovian analogue of the successor representation that carries information about the time required to reach a state for the first time rather than the full discounted occupancy of a policy.

**Summary Of The Review:**

The paper gives a thorough description of a variant of the successor representation and its potential application in a few different settings. Its main issue is that it frames itself as a sort of preliminary description of an idea, with a somewhat limited evaluation in any one setting, but very similar ideas have already been shown to scale to the types of problems commonly studied in contemporary deep RL. Including a comparison to these ideas (both experimentally and also in the framing) would improve the paper and situate it better in context.

---

> ### Author Response · Authors · 2021-11-16
> **Authors' Response**
>
>    Thank you for your detailed review and helpful advice! We're glad that you found our paper to be a thorough description of the FR and found FRP to be compelling. Please find a response to your concerns below.
>
>    1. *Framing*: We would first like to apologize and state that it was not our intention to portray the FR as the first instance of an idea relating to measuring the path length between states--this is a fundamental idea in RL and decision processes in general. Rather, our intention was to frame the FR as a representation which addresses an important form of non-Markovian structure that is not accounted for specifically by the SR. We agree that contextualization is important and have taken steps in the revision to remedy this--see the below points (and general responses above) for more detail.
>    2. *DG learning*: Thank you for the reference. This is an interesting paper of which we were not previously aware. We have added a discussion regarding the similarities and differences between the FR and DG learning to Appendix Section A.8 in the revision. One major difference that we note is that DG learning is a method for optimal control rather than evaluation--$DG(s,a,g)$ converges to the expected number of steps from $s$ to $g$ after taking action $a$ for $\pi^\star$. DG learning cannot be used to evaluate expected distances/discounts between states for an arbitrary policy. It is true that *for the optimal policy $\pi^\star$* $FR = \gamma^{DG}$ in deterministic environments (although not in stochastic environments), which is an interesting connection.
>    3. *DDL and TDMs*: Thank you for pointing these connections out! We were also not previously aware of these papers. We have since added references and discussion to the paper (see the general response above for a discussion of DDL/the DDF in particular; we discuss both DDL and TDMs in more depth in Appendix A.8). One important general point that applies to both of these approaches is that in order to estimate distances, they require a finite-horizon setting. This can lead to issues, for example, the conditioning problem discussed in the DDL paper. The FR avoids this by operating in the space of discounts.
>    4. *Experimental evaluation*: We have added two additional experimental results to the revision: 1) We have added DDLUS and a variant which we term DDLUS-G as baselines to the robotic reaching task (see the above note regarding DDL/the DDF for details, as well as an updated Figure 4 in the main text and Appendix A.3), finding that APF maintains its strong relative performance. 2) We have added an additional exploration task in the function approximation setting (see above and Appendix A.3), using the SF and FF as exploration bonuses in an analogous manner to the tabular settings in the original paper. We found that FFs performed strongly relative to SFs.
>    5. *Minor*: 1) Thank you! We've adjusted this in the revision. 2) In this case, we write $\mathbb E_\pi[\cdot|s_t]$ as shorthand for $\mathbb E_\pi[\cdot|s_t=s]$ to save space. We introduce this notation in Section 2 in the paragraph below Eq. 1, but we'd be happy to change it if it's unclear.
>
>    Thank you once again for your very helpful comments and suggestions. As a result, we believe that we've been able to strengthen the paper, particularly with regards to clarifying its context and improving our empirical evaluation. We hope that these changes have ameliorated your concerns. If so, we would be very appreciative if you might consider raising your score. We are also happy to discuss and address any additional concerns.

---

> > ### Comment · Reviewer_aieJ · 2021-11-22
> > **Reviewer response**
> >
> > Thank you for updating the paper! The contextualization after Definition 3.1, as well as the further discussion in Appendix A.8, are good additions.
> >
> > I still do think that, given that the paper exists in roughly the same space as DDL and TDMs, and because these papers had a more thorough experimental evaluation, it does raise the bar slightly for evaluation of this idea. The issue in the current state is that the proposed method is compared to DDL on only one task (and TDMs on none), and this particular task does not discriminate between the methods. (Their performance is close enough that you could expect hyperparameter tuning to change their relative ordering.)
> >
> > The relative performance is not a bad thing -- I don't want to suggest that FR / APF needs to outperform prior methods everywhere to be interesting! -- but I think it would be better if the choice of tasks gave us information about the tradeoffs between these approaches. Otherwise, they aren't very informative beyond saying that both methods pass a viability check. Is it possible to
> > 1. evaluate on the more standardized tasks from the DDL or TDM papers, for an apples-to-apples comparison?, or
> > 2. construct a continuous task in which you would expect the performances to differ more, to see whether the differences described in Appendix A.8 are reflected in experimental settings?
> >
> > As a more minor note, I am not sure why DG cannot be used for evaluation. The form described in Kaelbling 1993 looks like value iteration for finding $\text{DG}^*$, but I would think that you could simply replace the $\min_{a' \in \mathcal{A}}$ with an expectation over $\pi(\cdot \mid s')$ to evaluate a fixed policy $\pi$. Is there a reason this would not work?

---

> > > ### Author Response · Authors · 2021-11-23
> > > **Authors' Response**
> > >
> > > Thank you for your additional comments and suggestions! We're glad that you feel our additions strengthen the paper.
> > >
> > > To respond to your remaining points, we'd first like to respectfully disagree that our paper performs a less thorough experimental evaluation than the DDL and TDM papers. We test the FR and its extensions across a number of different use cases and settings, including exploration in both tabular settings and under function approximation on DeepSea, unsupervised RL in both MountainCar and a robotic reaching task, planning using a unique extension of the FR, and modeling animal behavior. Our experimental evaluation extends over a broader range of applications than these papers--and consists of more experiments overall than either, not that we think "number of experiments" is a valid measure of experimental rigor. This is not to say we don't think each of these papers aren't interesting and important, but simply that our focus is on the breadth of applications and *potential usefulness* of this representation, not the specific use case of unsupervised skill discovery with continuous control. We certainly agree that these papers perform a more thorough investigation of this particular setting, and that further experiments in this area would be valuable to readers especially interested in this area. By demonstrating the viability of our approach, this is exactly the kind of follow-up work we were hoping to inspire in the future. Unfortunately, we don't have time to run additional experiments by the revision deadline tonight, but we can certainly continue to perform experiments and report numerical results as part of further discussion. Below, you can find a table summarizing some basic differences between our approach and related methods, which we have also added to A.8 in the paper.
> > >
> > > |                            | FR/FF (ours) | SR/SF (Dayan 1993) | DDL (Hartikainen et al. 2020) | TDM (Pong et al. 2018) | DG (Kaelbling 1993) |
> > > | -------------------------- | ------------ | ------------------ | ----------------------------- | ---------------------- | ------------------- |
> > > | On- or Off-policy          | Both         | Both               | On-policy                     | Off-policy             | Off-policy          |
> > > | Evaluation or Control      | Evaluation   | Evaluation         | Evaluation                    | Control                | Control             |
> > > | Finite or Infinite Horizon | Both         | Both               | Finite Horizon                | Finite Horizon         | Finite Horizon      |
> > > | State representation?      | Yes          | Yes                | No                            | No                     | No                  |
> > >
> > > With respect to DG learning, we find this point slightly confusing. Replacing the $\max$ with an expectation is the same as shifting between $Q$-learning and TD learning, or shifting from the max-product algorithm to the sum-product algorithm. While they are closely related, they are different algorithms with different properties. This would certainly work, but it's not the method described in Kaelbling (1993). We'd also like to point out that DG learning, by learning distances rather than discounts, must be restricted to the finite horizon setting to guarantee convergence, and that $\gamma^{\mathbb E[k]} \neq \mathbb E[\gamma^k]$. We'd like to once again thank you for bringing these interesting works to our attention, and we think a comparison absolutely strengthens our paper. We do believe that these differences are more significant than the differences between the FR and SR, to which we devote the majority of our focus (and motivation).
> > >
> > > We sincerely thank you once again for helping us to strengthen our paper and for your insights and suggestions. If you believe these modifications constitute an improvement to the paper, we hope you might consider raising your score, and we will continue to work to add more experimental results for continuous control.

---

> > > > ### Author Response · Authors · 2021-11-29
> > > > **Additional Experimental Results in Unsupervised RL**
> > > >
> > > > In order to further strengthen our empirical evaluation of the FF applied to unsupervised RL, we
> > > > 1. Ran APF, APS, DDLUS, and DDLUS-G for 20 seeds on the DeepMind control suite Quadruped environment via the same Unsupervised RL Benchmark task set with the downstream "Quadruped-Run" task (a very similar set-up to the OpenAI Gym Ant environment used in the DDL paper).
> > > > 2. Ran an additional 10 seeds (bringing the total to 20) on the JACO reaching environment.
> > > >
> > > > We present our results below, measured after both 500k and 1M steps of fine-tuning. $\pm$ values denote 95\% confidence intervals, suggested by [1] to be a more reliable method for measuring the significance of empirical results in deep RL.
> > > >
> > > > | Task             | APF (Ours)  | APS         | DDLUS-G     | DDLUS     |
> > > > | ---------------- | ----------- | ----------- | ----------- | ----------- |
> > > > | JACO (500k)      | $215\pm 12$ | $199\pm 10$ | $190\pm 11$ | $169\pm 17$ |
> > > > | JACO (1M)        | $223\pm 6$  | $210\pm 4$  | $202\pm 8$  | $183\pm 10$ |
> > > > | Quadruped (500k) | $698\pm 15$ | $445\pm 27$ | $454\pm 26$ | $579\pm 43$ |
> > > > | Quadruped (1M)   | $851\pm 37$ | $483\pm 68$ | $521\pm 36$ | $810\pm 75$ |
> > > >
> > > > We can observe (1) on the JACO task, APF has overlapping intervals with APS after 500k steps, but no overlapping intervals after 1M steps, (2) on the Quadruped task, APF performs significantly better than other methods after 500k steps, and both APF and DDLUS show very similar performance at 1M steps. We also observe that APF attains the highest average performance in all cases.
> > > >
> > > > We believe these results lend further credibility to our contention that APF is a high-performing method for unsupervised RL. The results after 500k steps on Quadruped in particular support the idea that the vector-based FF representation provides a richer learning signal than the scalar-valued DDF, enabling faster learning. We also note that we kept the standard Gaussian sampling scheme for preference vectors $w$ in the pre-training phase for both APF and APS on the Quadruped task. Given the impact on downstream performance this change had on DDLUS vs. DDLUS-G in the JACO and Quadruped environments (itself, we believe, an interesting finding), modifying APF and APS to use the same max-distance intrinsic goal-setting strategy as DDLUS would likely lead to further improvements. Finally, we can observe the generally high degree of improvement of APF with respect to APS, the most similar baseline we consider.
> > > >
> > > > We hope that these additional results have further ameliorated your concerns. We'd like to reiterate that our intention with this paper was simply to introduce the FR/FF and demonstrate its use across a broad array of experimental settings (one of which being unsupervised RL) so as to inspire further work. We sincerely appreciate your time and your help in improving the paper. Thank you very much once again.

---

### Official Review · Reviewer_Ugir · 2021-11-04

**Correctness:** 3
**Technical Novelty And Significance:** 2
**Empirical Novelty And Significance:** 2
**Recommendation:** 6
**Confidence:** 4

**Main Review:**

This paper is well-written and well-motivated. The idea is simple but is a nice extension of the concept of the SR. It also seems promising on the different tasks considered by the authors. In particular, the case of non markovian rewards is studied in section 4.2.

How does the FR relate to the dynamic distance function introduced in https://arxiv.org/pdf/1907.08225.pdf ?

In Section 4.2, could you provide a reference for  “a bonus to maintain its effectiveness as time progresses in order to prevent the rate of policy improvement from exponentially decaying“? Unlike the analogy of SR / pseudo-count, it s still a bit unclear what the motivation for the FR is. Testing the benefit of the FR on harder exploration tasks would make the paper stronger I think (e.g., Machado et al 2020 tested their approach on Atari games)

Regarding infinite state spaces, the authors might want to note that recent works has shown that the SR can be extended the continuous state spaces when viewed as a measure without the need to rely on basis functions. See https://arxiv.org/abs/2101.07123 https://arxiv.org/abs/2103.07945

Minor points: Typos: take advantage *of*, theoeretical

**Summary Of The Paper:**

This paper proposes a new notion of state representation, the first-occupancy representation (FR), inspired by the Successor Representation. It is motivated by situations where the rewards are non-Markovian. Similarly to the SR, the FR can be learnt by TD learning. The usefulness of the FR is demonstrated on exploration, unsupervised RL, planning and escape behaviour tasks.


**Summary Of The Review:**

I think this paper provides a nice contribution to the topic of representation learning in RL. The authors provided theoretical results for their FR and demonstrated its benefits on 4 different tasks. Although it would be nice to confirm these findings on harder tasks e.g., in the case of section 4.1, I believe the variety of tasks considered by the authors still shows some promise which is why I recommend an accept for this paper.

---

> ### Author Response · Authors · 2021-11-16
> **Authors' Response**
>
>    Thank you for your detailed comments and helpful suggestions. We're glad you liked the paper and found the FR to be a promising idea! Please find a brief response below.
>
>    1. *The dynamic distance function*: Thank you for pointing this paper out! We refer you to the general response regarding dynamic distance learning above.
>
>    2. *Exploration*: Thank you for this, that was awkward phrasing. The meaning we intended was that we should prefer bonuses which don't decay over time regardless of the actual degree of exploration. We have amended the phrase to "a bonus to maintain its effectiveness over time" for simplicity. The core motivation for the FR as an exploration bonus is that its norm only increases when new states are visited (directly tied to exploration) or shorter paths are discovered, which indirectly supports exploration by reducing the effective distance between states. Finally, we refer you to the revised version of the paper (specifically, Appendix A.3), where we've added a new experiment testing the FF/SF in a similar manner to Machado et al. (2020) in the function approximation setting on a challenging task. We find that the FF bonus strongly outperforms the SF bonus on this problem.
>
>    3. *Infinite state spaces*: This is very interesting work which we were not aware of--we certainly think these are interesting ideas worth exploring in the future. We have noted these papers in the revision. Thank you for pointing this out!
>
>    4. *Typos*: Thank you! These have been addressed in the revision.
>
>    We hope this discussion and the associated additions strengthen the paper, and we thank you again for your time and consideration.

---

### Author Response · Authors · 2021-11-16
**Common Response**

We'd like to thank all the reviewers for their helpful feedback and advice on how to strengthen the paper. We are glad that the reviewers all found the FR to be a promising idea. In addition to the specific responses below, we've uploaded a new revision with the following salient changes:

- *Exploration*: We have added an additional exploration experiment in the function approximation regime, comparing DQN-style architectures with SF and FF reward bonuses on the DeepSea environment from [1]. We found that the strong performance of the FF carries over to this setting. The details are presented in Appendix A.3.

- *Unsupervised RL*: We have added additional baselines to the robotic reaching task, DDLUS [2] and DDLUS-G, finding that APF maintains its strong relative performance. We discuss these baselines in the note below on the dynamic distance function, as well as in Appendix A.3 (experimental details) and Appendix A.8 (discussion).
- *Related work*: We have added a new section discussing the connection between the FR/FF and related ideas in the literature (Appendix A.8). We also reference these ideas in the main text to better contextualize our contribution.
- *Minor*: We have moved the FRP pseudocode to the appendix due to space considerations. We have also fixed the typos pointed out by reviewers.

We believe that these changes strengthen the empirical evaluation of our method(s) and better situate it within the context of the literature. We hope the reviewers agree, and we thank you all once again for your thoughtful responses and consideration.

[1] https://openreview.net/forum?id=rygf-kSYwH

[2] https://arxiv.org/abs/1907.08225

---

### Author Response · Authors · 2021-11-16
**A discussion of the dynamic distance function**

We'd first like to thank Reviewers Ugir and aieJ for pointing out this interesting paper to us, as the *dynamic distance function* (DDF; https://arxiv.org/abs/1907.08225) is certainly related to the FR. Below, we discuss the similarities and differences between the DDF and the FR/FF, as well as describe additional experimental evaluations we carried out with the DDF in the unsupervised RL setting. **We have added this discussion to the paper appendix (Appendix A.8), as well as updated Figure 4 with the new results (details in Appendix A.3). We believe these additions strengthen the paper, and we hope the reviewers agree.**

**Discussion**

The *dynamic distance function* (DDF) is defined as
$$
d^\pi(s,s') = \mathbb E_{\pi} \left[\sum_{k=0}^{j-1} \gamma^k c(s_{t+k}, s_{t+k+1})\Big\vert s_t=s, s_j=s'\right],
$$
where $c: \mathcal S \times \mathcal S \to \mathbb R$ is a local cost function. In practice, $c(s_{t+k}, s_{t+k+1}) = \gamma= 1$, giving
$$
d^\pi(s,s') = \mathbb E_{\pi} \left[\sum_{k=0}^{j-1} 1 \Big\vert s_t=s, s_j=s'\right].
$$
In practice, $d^\pi(s,s')$ is parameterized via a neural network with parameters $\psi$ trained on-policy from full trajectories $\tau$ using the loss
$$
\mathcal L(\psi) = \frac{1}{2} \mathbb E_{\tau\sim\mathcal D,i\sim[0,T],j\sim[i,T]} (d_\psi^\pi(s_i,s_j) - (j-i))^2,
$$

where $\mathcal D$ is a buffer of stored trajectories. On a downstream navigation task, the agent policy is trained to minimize
$$
\mathcal L_\pi(\phi) = \mathbb E_{\pi} \left[ \sum_{t=0}^\infty \gamma^t d_\psi^\pi(s_t, g), \right]
$$
where $g\in\mathcal S$ is a goal state and $\phi$ are the policy parameters.

There are several important differences between the DDF and the FR, both in theory and practical application.

- This form of the DDF, as discussed by Hartikainen et al. (2020), is also naturally restricted to on-policy learning from full trajectories, and cannot be updated off-policy and/or via one-step temporal difference learning. A natural additional consequence is that it is only applicable to finite-horizon MDPs. This confinement to finite horizons is significant because it leads to the conditioning problem described by Hartikainen et al. (2020). This problem occurs because the DDF is conditioned on the policy successfully reaching the goal state--when this doesn't happen, it can lead to significant value estimation errors.

- A second difference is that the policy is trained, via Eq. (4), to minimize the *cumulative* discounted distance to the goal rather than via greedy distance minimization.

- Another difference is that Eqs. (1,2) don't explicitly require time step $j$ to be the first time the agent enters $s'$, although this may have been the authors' intention. In fact, this definition is more closely related to the SR than the FR (with equality in the infinite horizon setting), but the implementation of the DDF in practice is more closely related to the FR. It's also important to note that when $\gamma=1$, the number of steps $k$ appears linearly within the expectation rather than exponentially (this is important, as in general $\gamma^{\mathbb E[k]} \neq \mathbb E[\gamma^k]$.
- Another significant difference is that the DDF as implemented in practice cannot be in a meaningful sense be considered a state \emph{*representation*}, but rather a function mapping pairs of states to expected distances. That is, $d^\pi(s,\cdot)$ has no meaningful semantics when implemented as a mapping $d^\pi: \mathcal S \times \mathcal S \to \mathbb R$ and trained using Eq. (3). In practice, it is used to support policy optimization, rather than evaluation. In contrast, the FF representation, like the SF representation, maps a single state to a fixed length vector-valued encoding, $\varphi^\pi: \mathcal S \to \mathbb R^d$. This difference has significant implications for the applications of the DDF. In particular, it is unclear how the DDF could be used as an exploration bonus in the manner of the FF/SF, and the scalar value would make it challenging to implement the type of parallel updates useful for efficient planning.

---

> ### Author Response · Authors · 2021-11-16
> **DDF Experimental Evaluation**
>
> **Experimental Comparison**
>
> In light of the above discussion, we judged that the most appropriate domain for comparison between *dynamic distance learning* (DDL) and the FR/FF would be in unsupervised RL, in particular the Jaco robotic reaching task. As introduced by Hartikainen et al. (2020), we refer to the DDL in the unsupervised setting as DDLUS.  We used the same set-up for DDLUS as the original paper, with the exception that the base learning agent is DDPG, rather than SAC, to ensure consistency with the APS and APF models. In the original paper, the ```choose-goal``` function is implemented as $\mathrm{argmax}_{g\in\mathcal D} d_\psi^\pi(s_0,g)$, where $\mathcal D$ is a buffer of saved trajectories. In the pre-training phase, this encourages the agent to learn skills which reach/move as far as possible. Because we reasoned that this may not be an optimal choice for the Jaco task, we also implemented DDLUS with the same selection mechanism as used by APS and APF--that is, $g\sim\mathcal N(0, I)$--to ensure a more fair comparison. We termed this modified method DDLUS-G. We include the modified Figure 4 in our revision. We can see that APF outperforms both DDLUS and DDLUS-G, with the latter method outperforming the original DDLUS. We believe that APF's performance is likely a result of several factors, including that the vector-valued representation both acts as a richer substrate for learning (similar to hypotheses for the strong empirical performance of distributional RL (e.g., https://arxiv.org/abs/1902.08102, https://www.nature.com/articles/s41586-019-1924-6#Sec18 Supplementary Information)), as well as for policy evaluation in the fine-tuning regime.

---

> ### Comment · Area_Chair_BQ4A · 2021-11-23
> **Relationships to option-conditional prediction**
>
> Thank you for the detailed comparison to DDF.  I would like to know more on how the First occupancy Representation (FR) relates to option conditional predictions.  These predictions have been considered for similar reasons as pursued in this paper (as a representation and with policy evaluation), though perhaps not for the purposes pursued in this paper (exploration, Unsupervised RL).
>
> My understanding is that each element of the FR vector (from state $s$) is predicting the cumulant (pseudo-reward) signal $c=1_{s'}$ when following policy $\pi$ with termination probability $\beta=(1-\gamma)(1-1_{s'}) + 1_{s'}$ (alternatively expressed with the continuation probability of $\gamma(1-1_{s'})$).  This can be thought of as an option conditional prediction or a general value function (GVF).  The generic form of these predictions appeared in the option paper [1], developed into general value functions [2], led to GVF predictions that terminate on arrival at a state as above [3], and then a vector of GVFs used as a representation [4].  This form of predictive question also appears in option models [5,6] that share the reward factorization with SF.  The particular vectors in FR considered here are not in the form of an option model, but I believe they are a vector of GVFs as in [4].
>
> Separately, the FR aspirations have some similarity to replacing eligibility traces and first-visit Monte Carlo [7].  This might also be nice to comment on, though it is perhaps tangential.
>
> [1] Sutton, Richard S., Doina Precup, and Satinder Singh. "Between MDPs and semi-MDPs: A framework for temporal abstraction in reinforcement learning." Artificial intelligence 112.1-2 (1999): 181-211.
>
> [2] Sutton, Richard S., et al. "Horde: A scalable real-time architecture for learning knowledge from unsupervised sensorimotor interaction." The 10th International Conference on Autonomous Agents and Multiagent Systems-Volume 2. 2011.
>
> [3] Modayil, Joseph, Adam White, and Richard S. Sutton. "Multi-timescale nexting in a reinforcement learning robot." Adaptive Behavior 22.2 (2014): 146-160.
>
> [4] Schlegel, Matthew et al, "General Value Function Networks", Journal of Artificial Intelligence Research 70 (2021) 497-543
>
> [5] Sorg, Jonathan, and Satinder Singh. "Linear options." Proceedings of the 9th International Conference on Autonomous Agents and Multiagent Systems: volume 1-Volume 1. 2010.
>
> [6] Szepesvari, Csaba, et al. "Universal option models." Advances in Neural Information Processing Systems 27 (2014): 990-998.
>
> [7] Singh, Satinder P., and Richard S. Sutton. "Reinforcement learning with replacing eligibility traces." Machine learning 22.1 (1996): 123-158.

---

> > ### Author Response · Authors · 2021-11-25
> > **Authors' Response**
> >
> > Thank you for your interest in our work! We are glad that you found the DDF discussion useful. Indeed, the FR (and SR) can be seen as forms of option conditional predictions or GVFs. The form in which you have written the FR is quite nice in its expression of the termination condition in a Markovian fashion, and we will certainly add an appropriate discussion to the paper. We can further expand our discussion of the FR/FRP's connections to options in Appendix A.7.
> >
> > We also thank you for these interesting references! We've discussed several in the paper, but we were not aware of others. To briefly address several points:
> >
> > - **[3]:** In this case, the $i$th GVF is defined for cumulant $r^i$ and discount $\gamma^i$ as
> > $$
> > v_t^i \approx \sum_{k=0}^\infty (\gamma^i)^k r_{t+k+1}^i,
> > $$
> > which in this set-up can be written via a linear expression:
> > $$
> > v^i_t = \phi_t^\top \theta_t^i,
> > $$
> > where $\phi_t$ is the shared state-feature representation at time $t$, which in this case was a binary vector $[0,1]^d$ with each entry reflecting the presence or absence of a particular feature. That is, $\phi_t$ effectively functions as the *successor* features at time $t$, and where $\theta_t^i$ are learned preferences. These preferences are updated according to TD$(\lambda)$ with discount factors $\gamma^i$ and eligibility traces $\mathbf e_t^i$:
> > $$
> > \theta_{t+1}^i = \theta_t^i + \alpha(r_{t+1}^i + \gamma^i\phi_{t+1}^\top\theta_t^i - \phi_t^\top \theta_t^i)\mathbf e_t^i,
> > $$
> > where $r_{t+1}^i$ is the observed value of the given cumulant, which in this case was the reading of one of 53 sensors on a robot or the feature vector itself. It is rather unclear to us how in this case the GVF prediction terminates upon arrival at a state (or feature value). Rather, it seems to comprise a standard GVF across multiple time scales given by $\gamma^i$.
> >
> > - **[4]:** We thank you for this interesting and relevant example of the usage of GVFs as a form of state representation. In this instance, as in many other examples in the GVF literature, each GVF is learned with multiple discount factors in order to approximate learning over multiple effective horizons (e.g., the event "see red in 10 steps" is approximated by a "red" cumulant with discount factor $\gamma=0.9$—the effective horizon being $1/(1-\gamma) = 10$). The POMDP setting used in this paper with history-dependent policies parameterized by RNNs are especially interesting, as they enable the agent to learn non-Markovian representations, though an explicit "first-occupancy" concept isn't investigated. In this case, the approach is applied to time series datasets, without—as you note—evaluation in the exploration or unsupervised RL settings, or extension to planning.
> > - **Eligibility traces + first-visit monte carlo:** With respect to first-visit Monte Carlo, this is interesting because first-visit Monte Carlo only counts the observed (discounted) cumuluative reward *after* the first visit to a state, whereas the FR encodes the expected discount *until* the first visit. Eligibility traces are perhaps more similar in that they are are framed as "backward views" (Sutton & Barto, 2018 ch. 12).
> >
> > We thank you once again for drawing these interesting connections. We believe that the FR/FF combines a number of the desirable qualities of the approaches you have referenced (e.g., parallelism, vectorized state representations, evaluation across tasks) and leverages them in a variety of domains/application areas. Our goal was simply to formally introduce the FR/FF via a non-Markovian modification of the SR/SF and demonstrate its potential usefulness to the community as a basis for deeper domain-specific follow-up work. We will certainly add additional discussion to the paper! Please let us know if you have any other questions.

---

### Decision · Program_Chairs · 2022-01-20

**Decision:**

Accept (Poster)

**Comment:**

This paper introduces a first-occupancy representation for reinforcement learning problems, with potential benefits on problems with non-stationary rewards.  The representation is defined analogously to the successor representations, but captures the expected discounted time to first arrive at a state instead of measuring discounted visitations.  The paper develops the idea and illustrates some uses for exploration, unsupervised RL, and non-stationary reward functions (for example when food rewards are consumed).

The reviews brought forward a number of related older ideas in the literature, where several aspects of the method have been previously developed.  These include dynamic goal learning, option conditional predictions, general value functions, dynamical distance learning, and temporal difference models.  However, from the author response and ensuing discussion, the exact form of the proposed representation has not been studied for the purposes presented in this paper.  The reviewers appreciated the utility of this representation for problems with non-Markovian rewards, in particular that “the use of the first-occupancy values as an exploration bonus results in much more efficient exploration”.  Multiple reviewers commented on the desire for a stronger empirical evaluation, but they were satisfied with the contribution of the paper.

The reviewers arrived at a consensus that the paper contributes a new representation for RL problems with non-stationary rewards, with two reviewers strongly convinced and none opposed.  The paper is therefore accepted.